# Ensemble-Based Short Text Similarity: An Easy Approach for Multilingual Datasets Using Transformers and WordNet in Real-World Scenarios

Isabella Gagliardi *  and Maria Teresa Artese

Institute for Applied Mathematics and Information Technologies, National Research Council of Italy (IMATI—CNR), 20133 Milan, Italy; teresa@mi.imati.cnr.it
* Correspondence: gagliardi@mi.imati.cnr.it; Tel.: +39-02-23699487

**Abstract:** When integrating data from different sources, there are problems of synonymy, different languages, and concepts of different granularity. This paper proposes a simple yet effective approach to evaluate the semantic similarity of short texts, especially keywords. The method is capable of matching keywords from different sources and languages by exploiting transformers and WordNet-based methods. Key features of the approach include its unsupervised pipeline, mitigation of the lack of context in keywords, scalability for large archives, support for multiple languages and real-world scenarios adaptation capabilities. The work aims to provide a versatile tool for different cultural heritage archives without requiring complex customization. The paper aims to explore different approaches to identifying similarities in 1- or n-gram tags, evaluate and compare different pre-trained language models, and define integrated methods to overcome limitations. Tests to validate the approach have been conducted using the QueryLab portal, a search engine for cultural heritage archives, to evaluate the proposed pipeline.

**Keywords:** semantic textual similarity; pretrained language models; transformers; WordNet; QueryLab; ensemble methods

## 1. Introduction

The widespread availability of Web-based information highlights the importance of developing and promoting tools that can not only present search results but also seamlessly integrate this information while making suggestions to the user. Keywords have traditionally served as valuable indicators of textual content. However, their effectiveness depends on the ability to select terms that strike a balance between being broad enough to encompass multiple texts and specific enough to pinpoint cohesive subsets of data [1–3]. The reliance on different keywords, whether automatically extracted from the text or manually assigned by domain experts or catalogers, is a limitation that affects the performance of retrieval engines. Thus, situations may arise where keywords are synonymous or exist in different languages within the same text or dataset.

In this paper, a simple and easy-to-use yet effective approach to evaluate the semantic similarity of short texts, represented by keywords, is presented: given two sets of keywords from different sources and even in different languages, the method can perform the best matches between the lists, using both semantic and syntactic methods.

Semantic similarity algorithms, in general, rely on the transformation of strings of text into vectors. Transformers, Word2Vec, and GloVe, are all popular methods for this task, each with different strengths and use cases. In this paper, we prefer transformers for the following reasons:

(a) Pre-trained Models: there exist transformers language models pre-trained on large corpora, saving time and resources compared to training Word2Vec or GloVe embeddings from scratch, and more efficient than the pre-trained models.

(b)    Out-of-Vocabulary Handling: Transformers can handle out-of-vocabulary words better than Word2Vec or GloVe because they use sub-word tokenization methods (as used in this paper). This allows them to represent and work with words, not in the training data belonging to a niche context or domain.

(c)    Multi-lingual Support: Many pre-trained transformer models support multiple languages, making them suitable for multilingual NLP tasks. Word2Vec and GloVe models are typically language-specific.

We integrate semantic similarity approaches for context-free keywords with dictionary-based methods and string-based similarity. Different models and similarity measures are tested individually and ensemble to create an unsupervised, fully automatic approach that can be applied to different lists of terms (1-gram and n-grams), even in different languages.

The effectiveness and ease of use of this approach are attributed to the following key features:

- Unsupervised pipeline: Once hyperparameters and transformation models are optimized, the entire process becomes unsupervised. We use pre-trained models carefully chosen, eliminating the need to train the networks on specific data.
- Mitigation of the lack of context: the integration of different methods (neural network transformers, dictionary-based and syntactic) allows the mitigation of the lack of context of keywords and the definition of final similarity scores.
- Scalability: Using pre-trained language models, the approach can efficiently handle even archives with hundreds of elements.
- Multilingual support: The use of pre-trained multilingual text models enables the approach to efficiently manage archives containing documents in different languages, such as English, French, Italian, and others.
- Real-world scenarios: The experiments performed for this article demonstrate the ability of the method to adapt to real data without having to adapt it to a specific context. The use of ensemble methods makes it possible to overcome any critical problems that may arise due to unfamiliar words or different languages.

The proposed approach incorporates state-of-the-art tools, including transformers and pretrained language models, corpus-based (in this case, WordNet) and string-based similarity techniques. By synergistically integrating these tools, a versatile and adaptable tool has been created that provides a method suitable for various contexts.

The paper aims to define a simple yet effective method that is easily applicable to different cultural heritage archives and provides satisfactory results without requiring sophisticated methods to adapt to different cases. To achieve this, we set ourselves the following objectives:

(1)    Provide an overview of different approaches that can be adapted to identify overall similarities of 1- or n-gram tags.

(2)    Evaluate and compare the performance of different pre-trained language models on short texts/keywords, which are self-contained.

(3)    Define methods that can overcome limitations through integrated approaches after analyzing the results of individual methods.

To evaluate the pipeline, we conducted tests using data from the QueryLab portal [4], a search engine for archives on tangible and intangible cultural heritage capable of querying different datasets, both local and via web services simultaneously.

The paper is organized as follows: In Section 2, we briefly analyze the current state of the art. Then, in Section 3, we present our approach, highlighting its technical and innovative features, both individual measures and the overall approach. In Section 4, we present our experiments on QueryLab, where we thoroughly discuss and compare the results of individual approaches and the overall results in Section 5. This section also includes the presentation of results from applying our method to the WordSim353 gold standard dataset. Finally, Section 6 concludes the article with conclusions and future research.

## 2. Related Works

The study of word similarity is a fundamental task in natural language processing and has gained substantial attention from researchers. Numerous approaches have been proposed to measure the semantic similarity or relatedness between words. In this section, we introduce related works in word similarity.

There is a great number of survey works available in the literature regarding the similarity between words and phrases. In the comprehensive survey by Atoum et al. [5], the authors categorize similarity methods into two groups: word similarity and phrase similarity. Within each group, methods are further classified into three types: corpus-based, knowledge-based, and hybrid. Corpus-based approaches utilize statistical information from large text corpora and can be further divided into subcategories. Knowledge-based methods, on the other hand, rely on dictionaries or other structured resources to derive semantic knowledge. Hybrid methods combine elements from both corpus-based and knowledge-based approaches to leverage their respective strengths. For sentence-level similarity, context allows the incorporation of typical information retrieval (IR) features such as tf-idf or the utilization of large corpora like Wikipedia. Other authors, such as Gomaa et al. [6], Gupta et al. [7] or Sunilkumar et al. [8], in their surveys, categorize the similarity methods in analogous manners.

In [9], the authors conduct a systematic review of research on similarity measurement, analyzing the advantages and disadvantages of different methods. They categorize similarity measures into two major groups: those based on distance metrics and those based on text representation.

WordNet, known for its capability to represent semantic relationships, is widely used by researchers to measure semantic similarity. In [10], the authors provide a comprehensive review of various WordNet-based measures. They discuss the strengths and weaknesses of each approach in capturing semantic similarity. WordNet, in combination with a corpus, is also explored in [11], where a hybrid method is proposed using a novel similarity measure. This approach combines structural information from WordNet with statistical information from a corpus to enhance semantic text similarity.

Furthermore, the evaluation of semantic relatedness between lists of nouns using WordNet is investigated in [12]. The authors conduct experiments to evaluate the ability of different semantic relatedness measures, including latent semantic analysis (LSA), GloVe, FastText, and various WordNet-based measures, to predict differences in word recall between two lists of words.

Word similarity using word embeddings, such as Word2Vec or GloVe, has been a popular approach in natural language processing. These methods generate dense vector representations for words based on their co-occurrence patterns in large text corpora. Word similarity can be measured by calculating cosine similarity or other distance metrics between the respective word representations. Word2Vec and GloVe have shown promising results in capturing semantic relationships and syntactic regularities between words. In [13] the authors study whether similarity between short texts can be assessed using only semantic features. Vector representations of words, computed from unsampled data, represent terms in a semantic space in which the closeness of the vectors can be interpreted as semantic similarity.

In recent years, there has been a shift toward the use of transformer-based models, such as BERT (Bidirectional Encoder Representations from Transformers), for word similarity and related tasks [14]. The preference for BERT and transformer-based models in word similarity tasks stems from their ability to capture contextualized representations, take advantage of bidirectional language modeling, use large-scale pre-training, offer fine-tuning capabilities, and harness the power of transformer architecture. These advances have shown significant improvements in capturing word semantics and addressing the challenges of word polysemy and ambiguity.

In [15], the authors trace the evolution of semantic similarity methods from traditional NLP techniques, such as kernel-based methods, to the most recent research on transformer-based models, classifying them according to their basic principles as knowledge-based,

corpus-based, deep neural network-based and hybrid methods. Another survey using deep learning is presented in [16].

For the choice of which pretrained language model to use, comparisons made in specific fields, such as medical or financial [17,18], can be found in the literature. There are also works comparing the general models [19,20]. Their shortcoming is that these reviews and comparisons age very quickly. On the HuggingFace site [21] (from which the models used in this experimentation came) experimentation, for example, models are constantly and continuously added and updated.

To our knowledge, ours is the first approach to develop a method that relies on three different methods and their integration, entirely unsupervised in the "wild".

## 3. Materials and Methods

This paper aims to define a method to identify the most similar or related tags within two lists. As we will see later in the experimental part, the system can identify one or more terms that it judges to be similar, regardless of how similar they may appear at first glance. The system generates a sorted list of all compared terms based on similarity for each ensemble method. Figure 1 shows the pipeline of the approach, the purpose of which was found in the real case of integrating archives from different sources in QueryLab [4]. The pipeline involves steps or components designed to retrieve and present relevant terms based on their semantic similarity or association with the given word. The input to the system consists of two lists eventually containing synonymous, related, or corresponding terms that need to be identified. The various steps (identified as T1, T2 or T3) within the process generate intermediate results (R1, R2 or R3), which are subsequently utilized. For instance, during preprocessing, we obtain both the individual items for comparison and their associated languages. Based on this information, we select pretrained language models from a predefined list. WordNet-based similarity also considers language factors. For each item in the list, the final output of the method consists of one or more terms arranged based on the similarity computed by the chosen method or combination of methods.

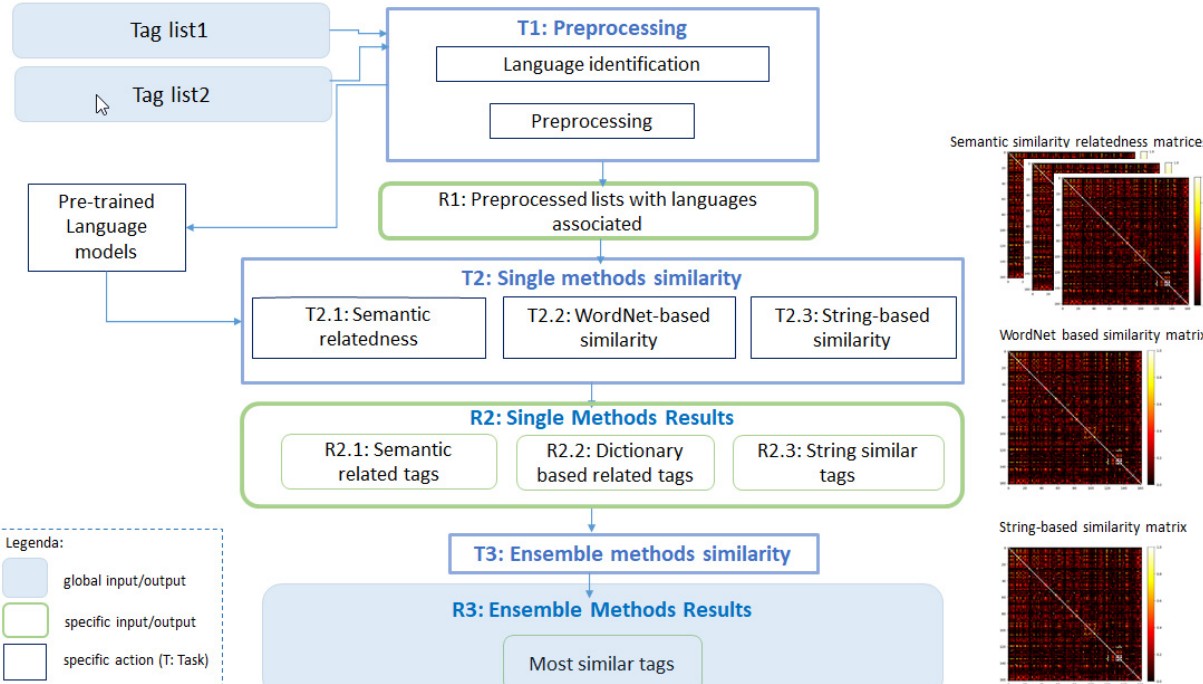

**Figure 1.** Schema of the proposed method. Similarity matrix: https://commons.wikimedia.org/wiki/File: Nuclear_Profile_Similarity_heatmap.png by Araoluwa3, CC BY-SA 4.0 <https://creativecommons.org/licenses/by-sa/4.0>, via Wikimedia Commons (last accessed on 15 September 2023).

### 3.1. Materials

One of the goals of the method is to define a tool capable of processing information in natural language in an unsupervised manner. The testing of the method was carried out on QueryLab. Several types of information from QueryLab were used to evaluate the effectiveness of the method, either list prepared by experts or keywords automatically extracted from QueryLab archives, as described in more detail below.

### 3.2. Language Identification and Preprocessing

Language identification and preprocessing are important steps in natural language processing (NLP) tasks (T1 in Figure 1). Language identification involves determining the language of a given text or document while preprocessing focuses on preparing the text data for subsequent analysis. Preprocessing is a crucial phase when working on data that involves applying a series of operations and transformations to prepare the data optimally for subsequent analysis. Preprocessing aims to improve data quality, reduce noise, eliminate irrelevant information, and make the data more suitable for machine learning algorithms or other analysis techniques.

### 3.3. Single Methods Similarity

Different similarity/relatedness techniques can be used to evaluate the similarity between terms with their strengths and weaknesses. In this paper, we have experimented with semantic similarity using pretrained Bert-like language models, dictionary-based and syntactic-based matching, reported in Figure 1 as T2.

**Semantic relatedness with transformers**: Transformers, as a remarkable development in the field of neural networks, have revolutionized the domain of natural language processing [22]. In contrast to conventional approaches that heavily depend on manually engineered features and statistical models, transformers employ a distinct mechanism known as self-attention [23]. This mechanism enables the model to dynamically allocate attention to various parts of the input, facilitating the capture of long-term dependencies within the language data.

Among the many pre-trained language models, BERT (Bidirectional Encoder Representations from Transformers) is one of the most influential and widely used [14]. BERT, one of the earliest pre-trained models, has had a significant impact on the field by providing a strong framework for learning representations. By training on large textual datasets, BERT can learn word embeddings that capture the meaning of words in context. The availability of BERT has greatly improved several natural language processing tasks, such as sentiment analysis, named entity recognition, and machine translation, leading to improved accuracy and performance.

Similarly, other models such as GPT, RoBERTa, and Mini-L6, also based on the transformer architecture, use similar techniques to capture contextual word meanings. These models support a wide range of natural language processing applications, each with unique enhancements designed to address specific NLP challenges. This contributes to the development and evolution of natural language processing, expanding its potential across domains.

These models are based on transformer neural networks and are trained on large amounts of unlabeled text data, enabling them to understand natural language patterns deeply. BERT and its counterparts from Microsoft, Facebook, OpenAI, and HuggingFace excel at bidirectional language modeling, which means they consider both preceding and following words to fully analyze word context. This approach allows a more nuanced understanding of word relationships and linguistic nuances. We conduct experiments to evaluate different pre-trained models.

In our approach, the process starts with pretrained models that represent individual words as vectors. These vectors encapsulate the semantic meaning of the respective words. However, when dealing with n-grams or phrases, additional techniques are required to combine the vectors associated with each component and generate a single vector representing the entire n-gram or phrase. This paper explores three distinct methods employed for this purpose, which are elaborated upon below. In the experimental section, we will present and analyze the results obtained by utilizing these methods.

One approach is to use the embedding of the [CLS] token, which is added to the beginning of each sentence in a batch during the pre-processing stage. The [CLS] token is designed to represent the entire sentence, and its embedding captures the meaning of the sentence. One way to compute the similarity between two sentences (in this case, n-grams) using pretrained language models is to take the dot product of their [CLS] token embeddings, which will give a score between $-1$ and 1, where 1 means the sentences are identical and $-1$ means they are completely dissimilar. Another way to compute similarity is by using the cosine similarity between the [CLS] token embeddings.

Another way to measure sentence similarity using BERT-like language models is to calculate the average of the token embeddings in each sentence. This is called the "mean-pooling" approach, denoted as [AVG]. To use this method for sentence similarity, we first run pretrained models on the input sentence to obtain the hidden states for each token. Next, we find the average of these hidden states for each sentence. Finally, we determine sentence similarity by calculating the dot product or cosine similarity of their average token embeddings. This approach can be beneficial in specific scenarios where we want to gauge the similarity between two sentences rather than solely comparing the [CLS] token embeddings. However, it's essential to acknowledge that this method may not be as effective as the [CLS] token embedding approach in all situations, as it might not capture the complete sentence meaning. This isn't a problem because we're working with very short phrases, typically consisting of just 3 or 4 words at most.

The last method for calculating sentence similarity using BERT is using the maximum value among the token embeddings for each sentence, commonly referred to as the "max-over-time pooling" approach and labeled as [MAX]. The process of computing sentence similarity using the max-over-time pooling approach follows the same steps as the previous method. We first obtain the hidden states for each token in the sentence using pretrained models. Then, we determine the maximum value among the hidden states for each sentence, specifically along the last dimension. Finally, we gauge sentence similarity by calculating the dot product or cosine similarity of their maximum token embeddings. This approach is suitable for identifying the most dominant meaning or feature within a sentence. However, it's important to note that, similar to the mean-pooling approach, this method may not capture the entire sentence's meaning. Additionally, it can be sensitive to outliers

**WordNet-based similarity**: WordNet is a lexical database and semantic network that organizes words and their meanings into a hierarchical structure [24,25]. It provides a comprehensive and structured resource for understanding the relationships between words, synonyms, antonyms, and the hierarchical structure of concepts. In WordNet, words are grouped into synsets (synonym sets), representing a set of words closely related in meaning. Each synset represents a distinct concept or meaning. Synsets are connected through semantic relations, such as hyponyms (subordinate concepts), hypernyms (superordinate concepts), meronyms (part-whole relationships), and holonyms (whole-part relationships).

WordNet's primary purpose is to facilitate the exploration of semantic relationships between words and to measure their semantic similarity. Again, we tested three measures to assess the similarity between n-grams:

1. The shortest path length measure computes the length of the shortest path between two synsets in the WordNet graph, representing the minimum number of hypernym links required to connect the synsets. This measure assigns a higher similarity score to word pairs with a shorter path length, indicating a closer semantic relationship. It will be referred to as a *path*.

2. Wu-Palmer Similarity: The Wu-Palmer similarity measure utilizes the depth of the LCS (Lowest Common Subsumer—the most specific common ancestor of two synsets in WordNet's hierarchy) and the shortest path length to assess the relatedness between synsets. By considering the depth of the LCS in relation to the depths of the synsets being compared, this measure aims to capture the conceptual similarity based on the position of the common ancestor in the WordNet hierarchy. It will be referred to as *wu*.

3. Measure based on distance: analogously to the shortest path length, this measure is also based on the minimum distance between 2 synsets. This measure, hand-crafted by the authors, considers that the shorter the distance, the greater the similarity. In this case, the similarity measure is calculated using this equation:

$$\text{min\_dist} = \frac{1}{\left(0.8 + \frac{\text{distance}}{4}\right)} \tag{1}$$

When distance > 0, in the other case min_dist = 1

This measure considers only the distance between synsets, in a depth-independent way, and was obtained by doing several tests and evaluations, so that much weight is given not only to synonyms (with distance 0) but also to hypernyms or hyponyms (distance 1) or siblings (distance 2). This measure will be referred to as *min_dist.*

**String comparison algorithms**: Jaro, Jaro-Winkler, Levenshtein and other similar similarity measures are string comparison algorithms that focus on quantifying the similarity (or distance) between two strings based on their characters and their order. These measures are useful in various applications, including record linkage, data deduplication, fuzzy string matching, etc. [26,27]. Here, it has been used Jaro-Winkler Similarity, which is an extension of the Jaro similarity measure. It incorporates a prefix scale that rewards strings for having a common prefix. The Jaro-Winkler similarity score ranges from 0 to 1, with 1 indicating a high similarity and a closer alignment of the prefixes.

*3.4. Ensemble Methods*

Since, in this paper, we have identified, defined, and implemented several measures to calculate similarity between objects, it is necessary to identify appropriate voting mechanisms to determine the best results (T3 in Figure 1). The starting point is to consider the following variables:

1. n pretrained language models: as we will see in the experimentation part, the datasets we tested our approach can be monolingual (English or Italian, so far) or multilingual (English, Italian or French, in the experiments). It is, therefore, necessary to identify the models that best represent the specificity of the data under consideration.

2. semantic relatedness: three different methods of calculating the representative vector to be evaluated must be compared.

3. WordNet-based similarity: again, there are three different ways of calculating the similarity between words.

Ensemble methods [28,29] aim to combine multiple individual models or methods to produce a more accurate and robust prediction or decision. It is based on the principle that the collective wisdom of diverse models tends to outperform any individual model in terms of accuracy, generalization, and stability. There are different voting schemes, such as majority voting, where the predicted class with the highest number of votes is selected, and weighted voting, where the models' predictions are weighted based on their performance or expertise. Weight-based and majority vote-based models play an important role in image classification, medicine, and manufacturing, enabling the prediction of failures and providing predictive support [30–34]. We used both methods in different phases to identify the more accurate results.

*3.5. Evaluation of the Results*

We are interested in evaluating the effectiveness and performance of each method when assessing the similarity between objects. By examining the outcomes, we can better understand how well these approaches capture and quantify the similarity between different objects, facilitating informed discussions and conclusions.

Borrowing ideas from information retrieval systems [1], a variety of metrics can be used to evaluate semantic similarity measures, including recall, precision, F1 scores, Dice and Jaccard coefficients:

1.  Recall: Recall measures the proportion of relevant items correctly identified or retrieved by a model. It focuses on the ability to find all positive instances and is calculated as the ratio of true positives to the sum of true positives and false negatives.
2.  Precision: Precision measures the proportion of retrieved items that are relevant or correct. It focuses on the accuracy of the retrieved items and is calculated as the ratio of true positives to the sum of true positives and false positives.
3.  F1 score: The F1 score combines precision and recall into a single metric. It is the harmonic mean of precision and recall and provides a balanced measure of a model's performance. The F1 score ranges from 0 to 1, with 1 being the best performance. It is calculated as 2 × (precision × recall)/(precision + recall).
4.  Dice coefficient: The Dice coefficient is a metric commonly used for measuring the similarity between two sets. Natural language processing is often employed for evaluating the similarity between predicted and reference sets, such as in entity extraction or document clustering. The Dice coefficient ranges from 0 to 1, with 1 indicating a perfect match. It is calculated as 2 × (intersection of sets)/(sum of set sizes).
5.  Jaccard coefficient: The Jaccard coefficient, also known as the Jaccard similarity index, measures the similarity between two sets. It is commonly used for tasks like clustering, document similarity, or measuring the overlap between predicted and reference sets. The Jaccard coefficient ranges from 0 to 1, with 1 indicating a complete match. It is calculated as the ratio of the intersection of sets to the union of sets.

These metrics provide different perspectives on the performance of semantic similarity measures. Recall focuses on the measure's ability to identify relevant similar pairs, precision emphasizes the accuracy of the identified pairs, and Jaccard similarity provides an overall measure of overlap between identified and reference pairs.

## 4. The Experimentation

The approach presented in this study comprises a series of sequential steps accompanied by detailed explanations and specific examples to provide a comprehensive understanding of the implementation process. Table 1 presents the pipeline of the approach.

*4.1. Datasets*

**QueryLab Platform**

The methods presented here have been designed with the idea of their use in the "wild", integrating them in QueryLab, a prototype dedicated to the integration, navigation, searching and preservation of tangible and intangible heritage archives on the web, with the aid of themed paths, keywords, semantic query expansion and word cloud [4].

QueryLab foresees different ways of querying inventories by collecting and managing data in a transparent way for the user. Archives are integrated in QueryLab in the most automatic way possible, both via web services and local ones. The datasets are steadily expanding and encompass archives related to intangible cultural goods and food, available in Italian and English. Additionally, they include data from prominent archives like Europeana (representing European countries), the Victoria and Albert Museum (UK), the Digital Public Library of America (DPLA, USA), and the Reunion des Musée Nationaux (RMN, France), among others.

**Table 1.** pipeline of the presented approach.

---

**# Task 1: Dataset Preparation**

- Harvesting process
- Language identification
- Preprocessing (possibly strip stopwords, accents, . . .)
- Output: items of interest

**# Task 2: Single Method similarity/relatedness**

- **# semantic relatedness**
- Choice of transformers and pre-trained models
- Fine-tuning of pre-trained Bert-like models to obtain the vectors
- Computation of similarity matrix using [CLS] tokens, means and max pooling
- Output: for each model, three lists of the most related tags, ordered by score value
- **# wordnet-based**
- For each tag1 in list1 and tag2 in list2:
- Synsets identification for tag1 and tag2, using language and automatic translation if necessary.
- Computation of the max path_similarity and wu_p_similarity
- Computation of the min distance and hand-crafted similarity
- Output: 3 lists of the most related tags, ordered by score value
- **# string-based similarity**
- Computation of Jaro-wrinkler similarity among a string
- Output: list of the most related tags, ordered by score value

**# Task 3: Ensemble similarity**

- **# ensemble for each pretrained model**
- Weighted Voting mechanism to create, for each language model, a unique list of most related tags
- Output: list of the most related tags, ordered by score value
- **# global ensemble**
- Majority Voting mechanism
- Output: lists of the most related tags, ordered by number of votes

**# Task 4: Evaluation**

- Ground truth creation
- Computation of recall, precision, and Jaccard similarity

---

Due to the diverse nature of the archives comprising QueryLab, the results obtained for the same query can exhibit significant variations. This discrepancy arises from keyword lists that may differ in form or language while maintaining semantic similarity. Consequently, there is a need to devise a method for harmonizing keywords, serving a twofold purpose:

- During the search phase, the query is to include all tags that surpass a predetermined similarity threshold. This expansion allows for a broader search scope, encompassing tags semantically similar to the original query. By including such tags, we aim to enhance the search results by considering related keywords.
- During the fruition stage, suggest elements that contain similar keywords to enhance the user experience. By identifying and recommending elements that share similar keywords, we provide users with relevant and related content. This approach enables users to explore and access information beyond their initial query, promoting a comprehensive and enriched browsing experience.

**Gold standard datasets**

In addition to the data collected, the evaluation of the quality of the approach is based on two gold standards for word similarity: WordSim353 [34,35] and SimLex999 [35,36].

WordSim353 is a widely used benchmark dataset for evaluating word similarity and relatedness. It consists of 353-word pairs, and each pair is assigned a similarity score by human annotators based on their perceived similarity or relatedness. The dataset covers a range of semantic relationships, including synonyms, antonyms, hypernyms, and more. The WordSim353 dataset is a standard evaluation resource for measuring the performance of word embedding models and other techniques in capturing semantic similarities between

words. Researchers often use this dataset to assess the quality of their models and compare their results with other approaches.

SimLex-999 is another widely used benchmark dataset for evaluating word similarity and relatedness. It consists of 666 noun-noun pairs, and each pair is assigned a similarity score by human annotators based on their perceived similarity. The dataset covers a diverse range of semantic relationships and includes words with different levels of similarity and relatedness.

### 4.2. Dataset Preparation

**Language identification and preprocessing**

In many of the processing and similarity evaluation tasks, language plays a crucial role. While certain datasets used in our experiments have known languages, such as benchmark datasets or Italian recipes, the data collected through automated methods often lacks fine-grained language information. To address this, we perform preprocessing to extract detailed language information from the data [37].

It is important to note that the preprocessing step is only applied to data collected in QueryLab archives. Since we are already working with keywords, the main activity performed is grouping tags within each dataset based on specific criteria. Specifically, we group tags with a distance of 0 in WordNet and a Jaro similarity score greater than 0.95.

### 4.3. Single Methods Similarity

The assessment of similarity in this experiment emphasized several key aspects:

1. Determining the best similarity measure: Various similarity assessment methods were utilized to calculate similarity scores between tags. The evaluation aimed to identify the similarity algorithm that best captured the semantic similarity between the given tag and other tags in the datasets.
2. Identifying the single tag with the highest similarity: Based on the calculated similarity scores, the tags that exhibited the highest similarity to the given tag were identified, one for each similarity measure.
3. Identifying the three most similar features: In addition to finding the single tag with the highest similarity, the evaluation process also focused on identifying the three most similar features. These features are the elements (1-gram or n-grams) that demonstrated high similarity to the given tag.

Considering these three aspects, the similarity process aimed to provide a comprehensive assessment of the semantic similarity between tags.

**Semantic relatedness**

For evaluating semantic similarity (T2.1 in Figure 1), we first convert tags, which can be 1-gram or n-grams, into vectors using pretrained language models specifically designed for "sentence similarity" tasks. The availability of pretrained models is extensive and continuously expanding.

One of the aims of the paper is to compare how different pretrained models perform in the evaluation of semantic similarity: we tested 2 or 3 different language models in various languages. We select these models based on information gathered from literature sources or platforms such as Hugging Face and models developed by major tech companies like Microsoft, Facebook, or Google. We utilize three sets of pretrained models, as described in Table 2.

The initial step in evaluating semantic similarity involves generating similarity matrices. The similarity matrix requires a single vector for each element being compared. When utilizing a BERT-like model, we employ specific tokens, such as [CLS], which represents the entire sentence, or the average of tokens [AVG], or the [MAX] as mentioned previously. The input text undergoes preprocessing, involving tokenization and encoding into numerical vectors. These vectors are then fed into the transformation model, which produces contextualized embeddings for each token in the input text.

**Table 2.** Pretrained language models tested for English, Italian and multilingual datasets.

| No. | Model Name | Pretrained Language |
|---|---|---|
| 1 | 'sentence-transformers/paraphrase-MiniLM-L6-v2' | English |
| 2 | 'flax-sentence-embeddings/all_datasets_v3_mpnet-base' | English |
| 3 | 'tgsc/sentence-transformers_paraphrase-multilingual-mpnet-base-v2' | English |
| 4 | 'nickprock/sentence-bert-base-italian-xxl-uncased' | Italian |
| 5 | 'tgsc/sentence-transformers_paraphrase-multilingual-mpnet-base-v2' | Italian |
| 6 | 'LLukas22/paraphrase-multilingual-mpnet-base-v2-embedding-all' | Multilingual |
| 7 | 'tgsc/sentence-transformers_paraphrase-multilingual-mpnet-base-v2' | Multilingual |

To construct the similarity matrix, the [CLS], [AVG] or [MAX] tokens are compared pairwise using a distance metric, typically cosine similarity. The resulting scores reflect the similarity relationships between the [CLS], [AVG] or [MAX] tokens within the input text, allowing for the creation of the similarity matrix.

**WordNet-based similarity**

The approach for WordNet-based similarity (T2.2 in Figure 1) is based on the principle that words with greater similarity exhibit smaller distances within the WordNet structure. However, the search for tag synsets in WordNet is language-dependent. We began with two key considerations:

1. Not all WordNet synsets have been translated into all languages. This implies that some synsets may not be available in certain languages, which can affect the matching process.
2. By employing machine translation on tags, we can increase the likelihood of finding a corresponding synset. This strategy helps overcome language barriers and enhances the chances of locating relevant synsets.

For tags comprising multiple words, if a word's lemma (base form) is not found directly, we split and identify the lemmas and synsets of individual words. This allows for a more granular analysis and matching process.

Similar to the previous approach, a similarity matrix is generated using the three measures described above. These scores are the basis for constructing the similarity matrix, facilitating further analysis and evaluation.

**String-based similarity**

String-based similarity techniques (T2.3 in Figure 1) are valuable for detecting minor script changes, small errors, or variations such as singulars and plurals, which can hinder effective searches across archives or data from different sources. By utilizing string-based similarity, we can address these issues and enhance the search process.

*4.4. Ensemble Methods Similarity/Relatedness*

Various similarity assessment methods using different characteristics were employed in this experiment, making it necessary to take an overall approach that provides a comprehensive and balanced assessment of similarity, considering multiple measures and their respective strengths in the overall assessment process. The ensemble evaluation approach followed the path of individual evaluation and the best three results. This led to using the weighted voting mechanism on the first best result (with the maximum similarity value) among the three, either considering all methods overall or based solely on semantic similarity or WordNet.

We defined ensemble methods by considering a combination of methods, language models and frequency of the most similar terms:

1. Single Model Ensemble: This ensemble approach considered the best similarity score for each pre-trained model by combining the results of the various similarity evaluation techniques. Using the weighted voting mechanism (see Table 3a, rows 14–17), the

ensemble evaluation aimed to capture the overall similarity between the tags while considering the specific evaluation techniques employed. On the other hand, using the majority voting techniques (see Table 3b, rows 1–4), we can consider the consensus among various approaches.

2. All Models Ensemble: In this case, the ensemble is measured by pooling the results from all the language models considered. In this case, greater importance is attributed to the identified term than the similarity value, thereby expanding the range of similar terms, including those with some level of relatedness. What is taken into consideration is the frequency of term selection (Table 3b, rows 5–8).

3. Most Frequent Terms Ensemble: The combined results considering all pre-trained models, with the majority voting mechanism applied only to the most frequent terms, capture the collective decision of multiple language models (Table 3b, rows 9–12).

**Table 3.** (a) single similarity method result label. (b) ensemble similarity method result label.

| | | (a) single similarity method result label | |
|---|---|---|---|
| 1 | s1s.[CLS] | Single method using semantic similarity, taking the most similar element and respectively [CLS], [AVG] and [MAX] | individual similarity |
| 2 | s1s.[AVG] | | |
| 3 | s1s.[MAX] | | |
| 4 | s1w.path | Single method using WordNet-based similarity, taking the most similar element and respectively path, wu and min_dist | |
| 5 | s1w.wu | | |
| 6 | s1w.min_dist | | |
| 7 | s.Jaro | Single method using Jaro similarity | |
| 8 | s3s.[CLS] | Single method using semantic similarity, taking the most three similar elements and respectively [CLS], [AVG] and [MAX] | |
| 9 | s3s.[AVG] | | |
| 10 | s3s.[MAXS] | | |
| 11 | s3w.path | Single method using WordNet-based similarity, taking the most three similar elements and respectively path, wu and min_dist | |
| 12 | s3w.wu | | |
| 13 | s3w.min_dist | | |
| 14 | e1.model | Single method ensemble, using weighted voting mechanism respectively on the most and most three similar elements obtained by all single methods (for the specific language model) | Single model ensemble (see Section 4.4 point 1) |
| 15 | e3.model | | |
| 16 | e1s.model | Single method ensemble, using weighted voting mechanism respectively on the most and most three similar elements obtained by semantic methods (for the specific language model) | |
| 17 | e3s.model | | |
| | | (b) ensemble similarity method result label | |
| 1 | e1.model | Single methods ensemble, using majority voting mechanism respectively on the most and most three similar elements obtained by all single methods, semantic (s) or WordNet-based algorithms (for the specific language model) | Single model ensemble |
| 2 | e3.model | | |
| 3 | e3s.model | | |
| 4 | e3w.model | | |
| 5 | e1.all | As mentioned above, combining the results obtained by each language model | All Models Ensemble(see Section 4.4 point 2) |
| 6 | e3.all | | |
| 7 | e3s.all | | |
| 8 | e3w.all | | |
| 9 | e1.max | As above, combining the results obtained by each language model, applied only to the most frequent terms | Most Frequent Terms Ensemble (see Section 4.4 point 3) |
| 10 | e3.max | | |
| 11 | e3s.max | | |
| 12 | e3w.max | | |

**User ground truth**

To evaluate the quality of the approach, it is crucial to have a "ground truth" dataset for comparing the results and calculating metrics such as recall and precision. For this purpose, a procedure involving annotators, including experts in the Cultural Heritage field and ordinary individuals, was established.

In the annotation process, the annotators were tasked with identifying the word or words most similar to a given target word. This task allowed for the identification of terms that closely matched the target word's meaning. However, in cases where finding suitable terms proved challenging due to a lack of similar options, the annotators were provided with the following choices:

- Eliminate the word: If no suitable term could be found, the annotators could exclude the word from the annotation process due to the unavailability of appropriate alternatives.
- Make an extremely bland association: In cases where the annotators faced significant difficulty finding similar terms, they could make a less precise or less contextually relevant association. This option aimed to capture any resemblance or connection, even if it was not an ideal match.

In the instructions for annotators, the first choice is preferred. To assist the annotators in this task, the system provided suggestions that exhibited very high Jaro similarity values or had a WordNet distance of less than 2. These suggestions were intended to guide the annotators and help them identify potentially relevant terms.

In the annotation process, the annotators were tasked with identifying the most similar tag and the three most related or similar tags in some way to the given target tag. This approach aimed to capture a broader range of related terms and expand the understanding of semantic relationships. By requesting the annotators to identify the three most similar or related tags, the evaluation process went beyond a single best match. It encompassed a small set of related tags that shared semantic similarities or connections with the target tag. In this first phase of evaluation, we have employed a limited number of evaluators with diverse backgrounds.

## 5. Results and Discussion

### 5.1. Datasets Used

The data collection process was based on queries to the QueryLab archives, using the REST API protocol at the source and storing the following data for the first n results. Each archive organizes the search results differently and, therefore, requires ad hoc procedures.

List similarity experiments were performed on the following data:

1. Keywords extracted dynamically in QueryLab. The interesting thing is that QueryLab's archives are multilingual, and almost all of them can respond with English terms. When tested, however, different languages coexist, e.g., RMN or DPLA. Here are the results of the queries:

    (1) *mariage* on Europeana and RMN, using respectively 120 and 100 items;
    (2) *wedding* on Victoria & Albert Museum and DPLA., using 100 items for both archives.

An initial analysis of the data, shown visually in Figures 2 and 3, shows that there are indeed clearly identifiable synonyms and that the language of the words is different. These two datasets were chosen because they possess certain characteristics that make them interesting. The *Mariage* results consist of lists of terms in English and French, whereas the *Wedding* results contain only English tags, using synonyms. Therefore, the chosen pre-trained models are multilingual for *Mariage* to handle terms in English and French, whereas English-specific pre-trained models were used for *Wedding*.

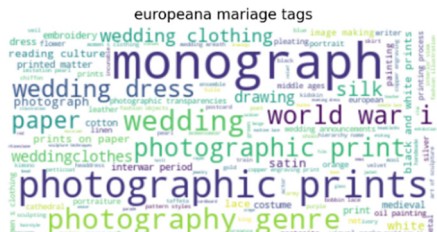
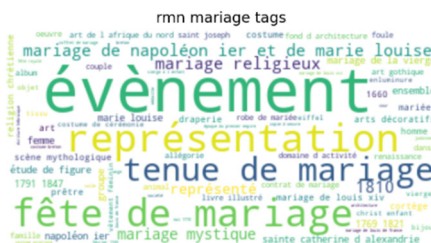

**Figure 2.** word cloud for *Mariage* on Europeana and RMN.

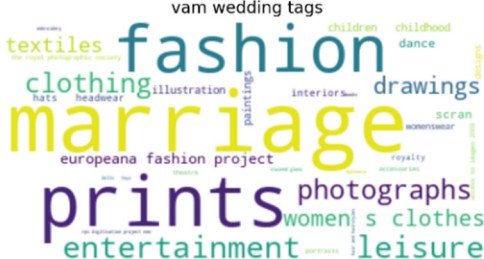 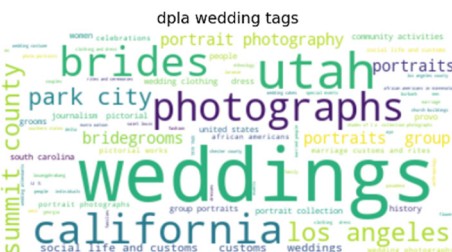

**Figure 3.** word cloud for *Wedding* on Victoria & Albert Museum and DPLA.

2.  Intangible heritage-related tag lists extracted from QueryLab. These lists are hand-built by experienced ethnographers. The tags we compare are those defined for the Archives of Ethnographies and Social History AESS and its transnational version for the inventory of the intangible cultural heritage of some regions in Northern Italy, some cantons in Switzerland and some traditions in Germany, France and Austria [38] and those of UNESCO [39], imported in QueryLab. The language is the same (English), but there are extremely specialized terms, making it difficult to assess similarity. This dataset is called *ICH_TAGS* and comprises 300 elements for both lists.

3.  tag lists referring to cooking and ingredients, in Italian, again taken from QueryLab. The interest is to handle data belonging to a specific domain in a language other than English, both for semantic and dictionary-based similarity. The pretrained models used are those for Italian. A comparison was made with a multilingual model. In the following, called *Cook_IT*, it is composed of 100 and 300 items.

4.  WordSim363, the gold standard.

*5.2. Results Evaluation*

The performance of the approach on different datasets has been evaluated, and the results were summarized in graphs displaying the recall, precision, F1, Dice and Jaccard scores. The numerical data used to create these graphs is presented in Appendix A.

To enhance clarity, the figure labels have been included in Table 3. The individual similarity methods are labeled in Table 3a, while the ensemble results are presented in Table 3b. The methods listed in Table 3 correspond to distinct elements in the graph, each associated with a specific language model. However, for the first four methods in Table 3b, the number of occurrences in the graph corresponds to the number of pretrained models used: two for Italian and multilingual and three for English.

***Mariage* dataset on Europeana and RMN archives**

Figures 4 and 5 report evaluation results on *Mariage* from Europeana and RMN, described in Section 5.1. It should be noted that Europeana is a multilingual archive, while RMN primarily features French content with some metadata available in English. We identified two pretrained models specifically trained on multilingual datasets for our approach, employing automatic language detection.

In Figure 4, the depicted outcomes correspond to the single similarity methods as previously defined. Specifically, for the semantic similarity methods, the results pertain to the two language models used for multilingual texts. The figure presents the results obtained by combining the individual methods using a weighted voting strategy applied to the methods within the current model.

Our primary focus is on achieving high recall, indicated by the red color in the graph, considering the objective of the approach. When examining single semantic models, where only the first result (s1s) is considered, the recall is comparatively lower than WordNet-based methods (s1w). This observation holds even considering the top three results (s3s and s3w). As expected, string-based similarity yields very low results, and it is utilized in ensemble methods to reinforce similarity when it surpasses a specified threshold, empirically set at 0.9.

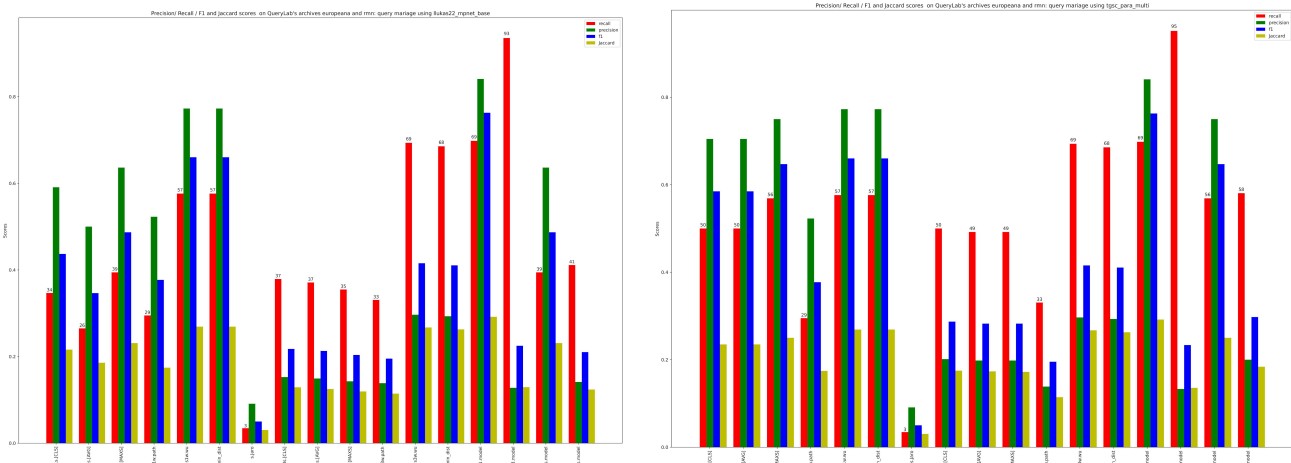

**Figure 4.** Single methods results for *Mariage*. Labels are those of Table 3a.

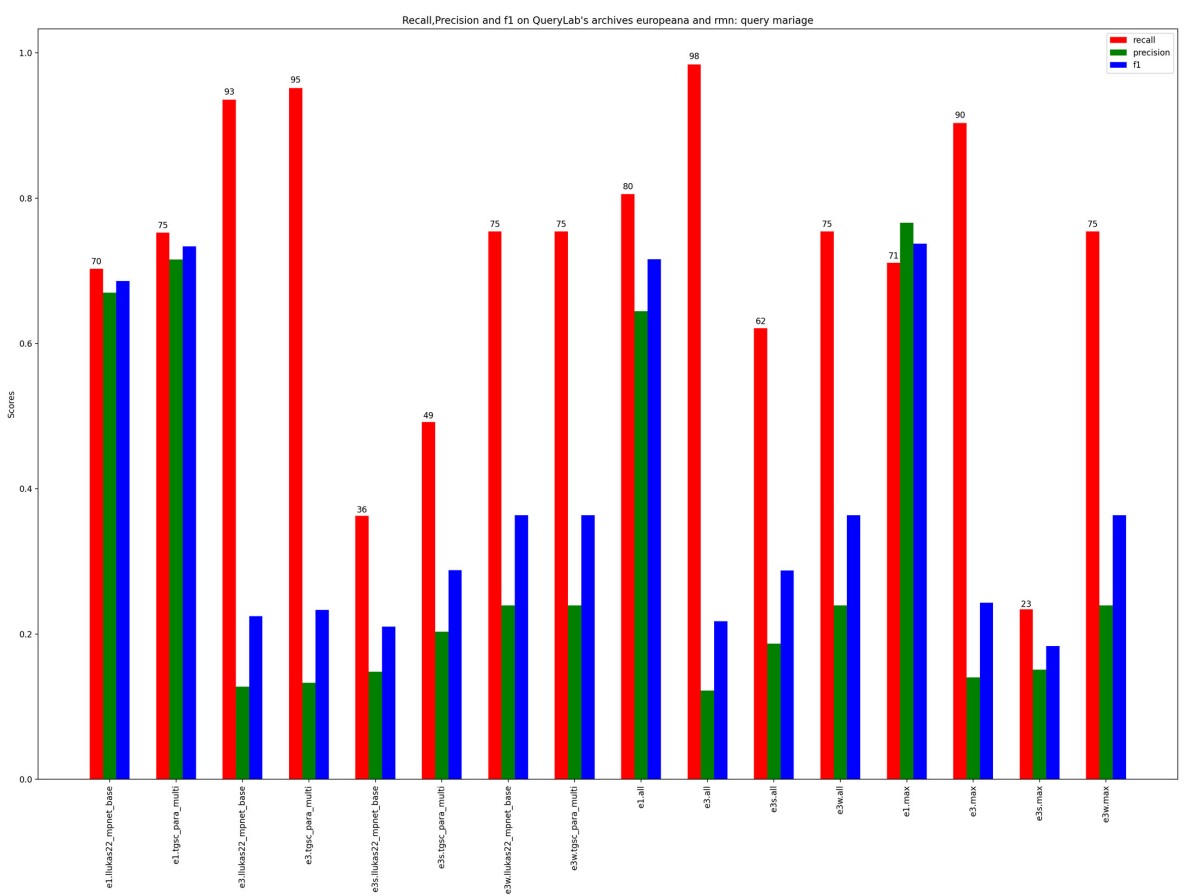

**Figure 5.** ensemble methods result for *Mariage*. Labels are those of Table 3b.

The last four columns of Figure 4 obtained integrating the results using ensemble methods on a single language model, the recall for considering all methods applied to the first three results achieves a promising 93 (e3.model), surpassing both the recall obtained by solely considering the term evaluated as most similar through majority voting (recall 69 with e1.model) and employing only semantic methods. The latter approaches yield a recall of 39 and 41 for a single term (e1s.model) and the first three (e3s.model), respectively.

Figure 5 presents the ensemble results for different scenarios, including all methods or limiting only to semantic or WordNet-based methods. The highest recall is obtained using a result composition method that considers all pre-trained models and takes the best three

results (e3.all). This result significantly improves the scores obtained by semantic and/or WordNet-based methods alone and by taking only the best result. We do not observe much variation in the use of different pre-trained language models.

### *Wedding* **dataset on DPLA and Victoria & Albert Museum archives**

Figures 6 and 7 show the results obtained on the wedding query in the DPLA and Victoria & Albert Museum archives. In this case, the tags were identified in English, and the pretrained language models for that language were used. The results obtained replicate those for *Mariage*. Again, the ensemble methods, both for the single model and overall on all models, outperform the single methods. The solution of taking the first three results yields a recall of 98 percent with the e3.model in both models (Figure 6). This value reaches 100% recall in e3.all.

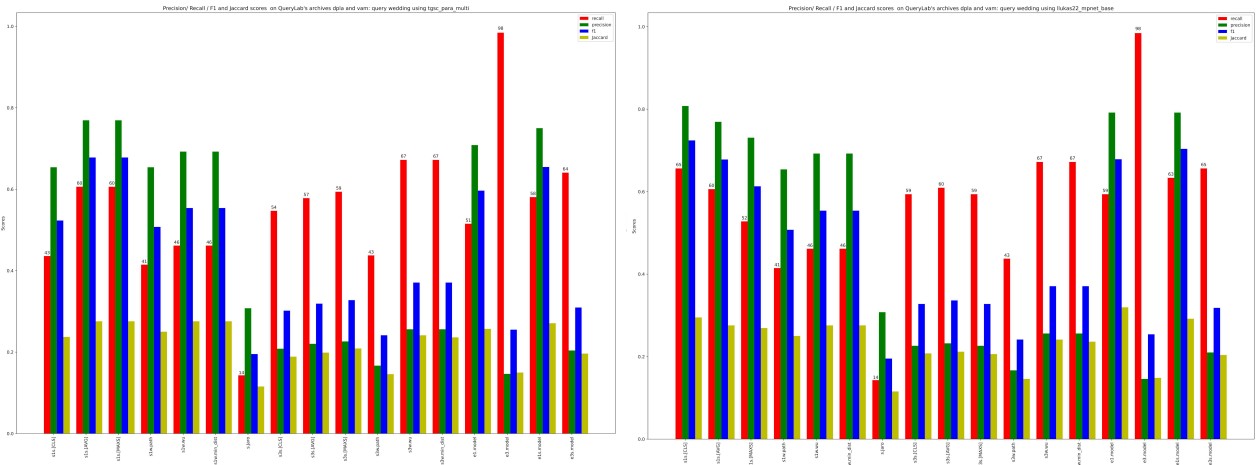

**Figure 6.** Single methods result for *Wedding*. Labels are those of Table 3a.

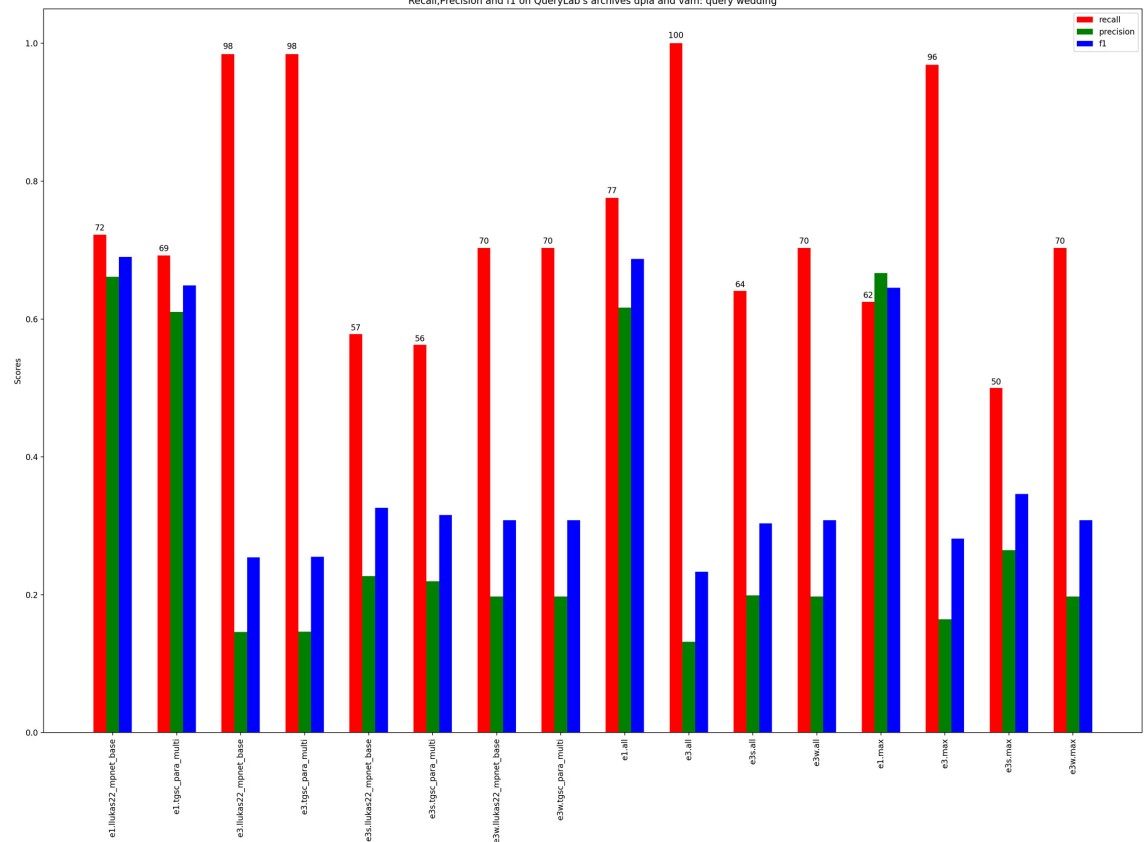

**Figure 7.** Ensemble methods result for *Wedding*. Labels are those of Table 3b.

*ICH-tags* **dataset**

The distinctive feature of ICH tags is their utilization of highly specific terms. The potential challenge posed by classical word embeddings, which could result in out-of-vocabulary (OOV) terms, is effectively addressed through the tokenization process of transformers. As a result, the similarity between terms is accurately computed, even in cases where the model is unfamiliar with individual words, thanks to the incorporation of tokenizers and fine-tuning.

As shown in Figures 8 and 9, the previous results are confirmed here: the best single model result is from e3.model, and the best ensemble methods result is e3.all.

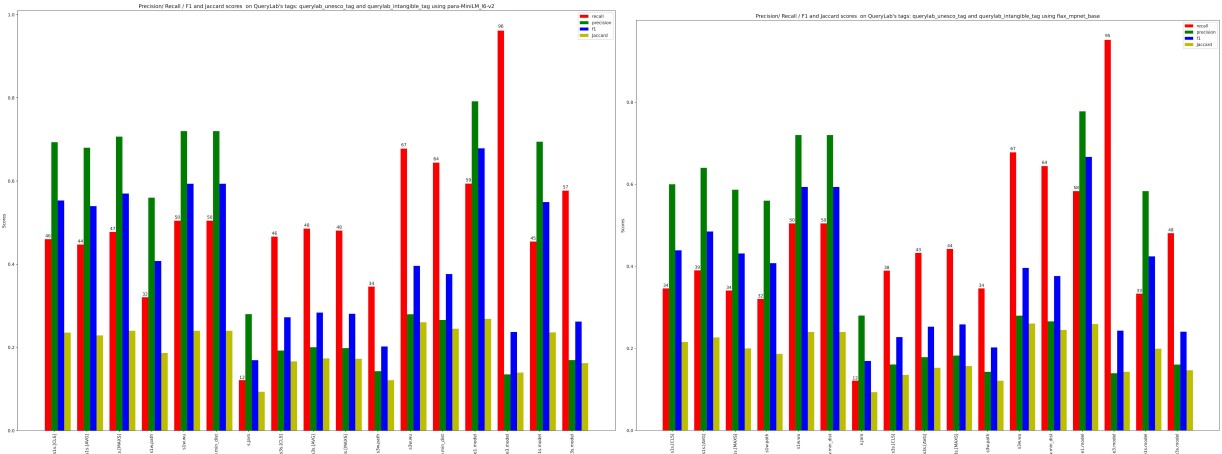

**Figure 8.** Single methods result for *ICH tags*. Labels are those of Table 3a.

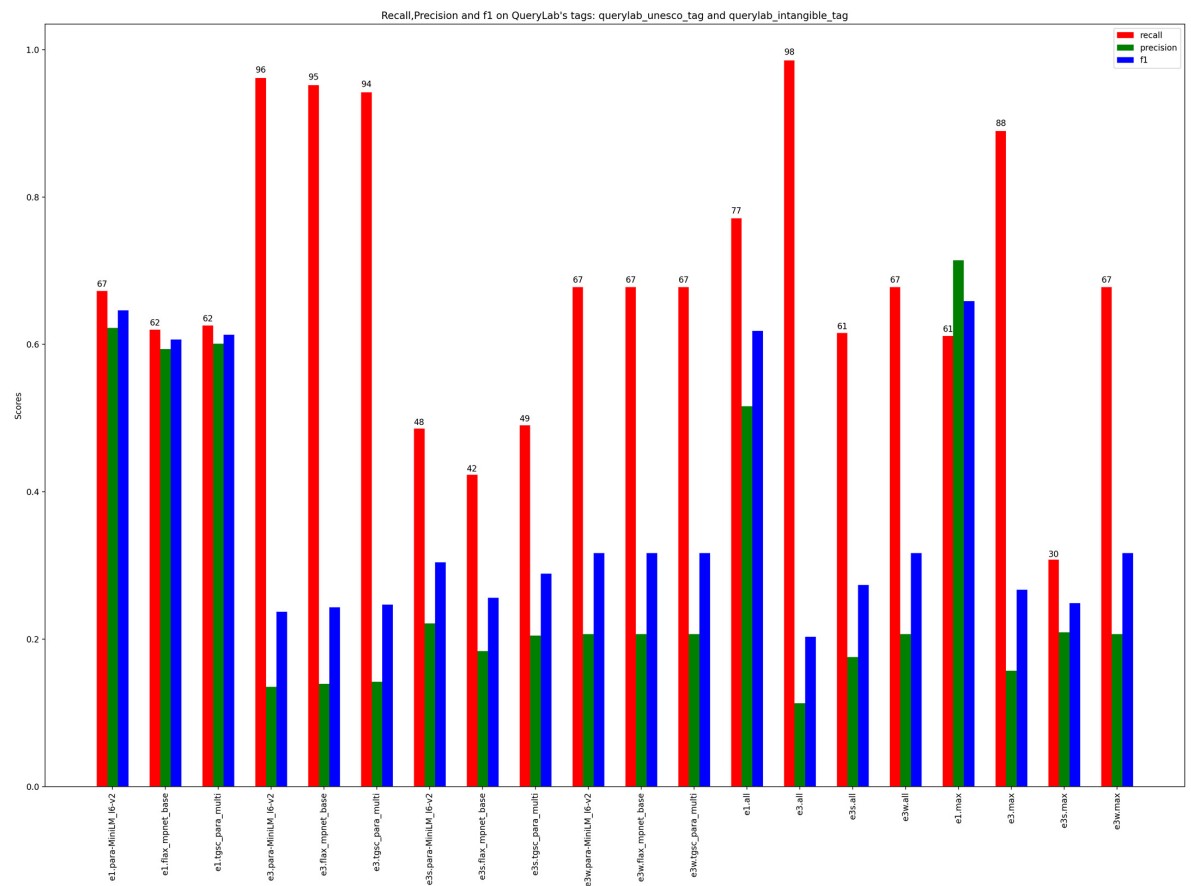

**Figure 9.** Ensemble methods result for *ICH tags*. Labels are those of Table 3b.

*Cook_IT* **dataset**

Figures 10 and 11 pertain to the Italian dataset comprising terms associated with recipes and ingredients used in Italian cuisine. These terms are commonplace within the specific domain but not typically in a general-purpose lexicon. However, by employing a tokenizer, this challenge is effectively addressed, enabling the measurement of similarity between the two sets with promising outcomes.

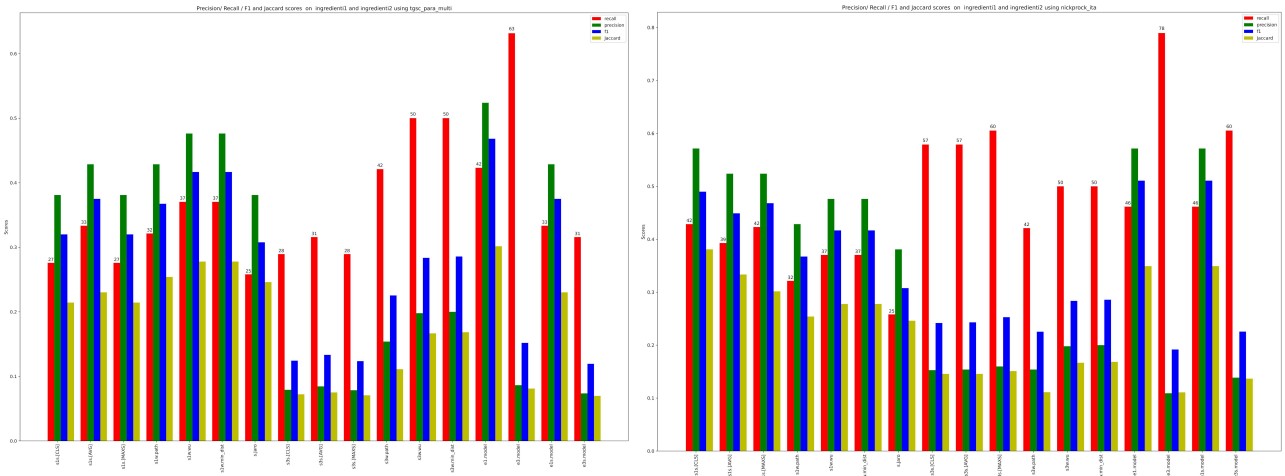

**Figure 10.** single methods result for *Cook_IT*. Labels are those of Table 3a.

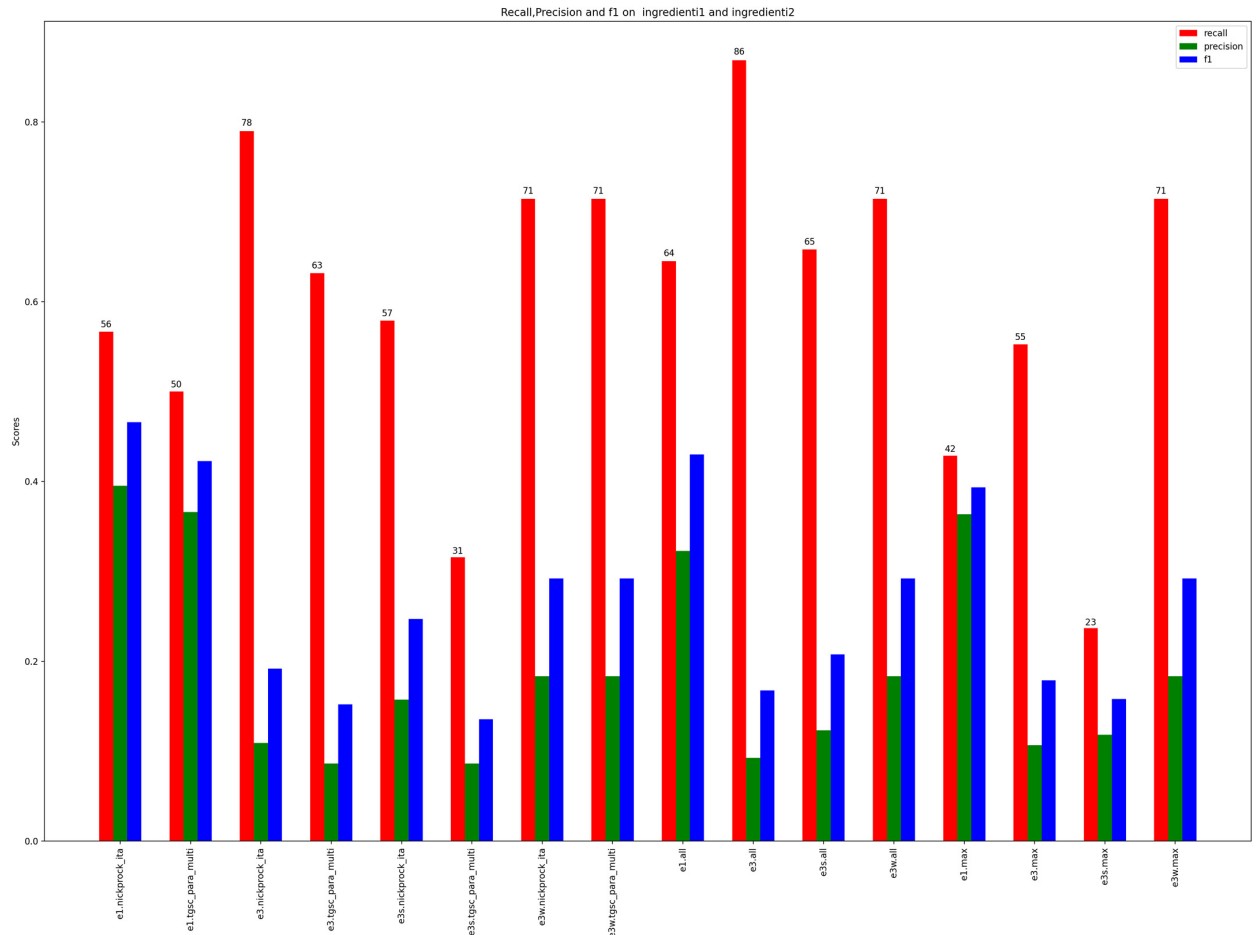

**Figure 11.** ensemble methods result for *Cook_IT*. Labels are those of Table 3b.

*Gold standard WordSim353 dataset*

Figures 12 and 13 are for the gold standard dataset. In this case, the recall is very high for most of the single methods. Once again, it is observed that ensemble methods focusing on the first three results demonstrate superior performance. This finding applies to individual models (Figure 12, e3.model) and ensemble methods that combine multiple methods and language models (Figure 13, e3.all). In both cases, considering the first three results leads to improved outperforms compared to considering only the most similar term or other combination mechanisms.

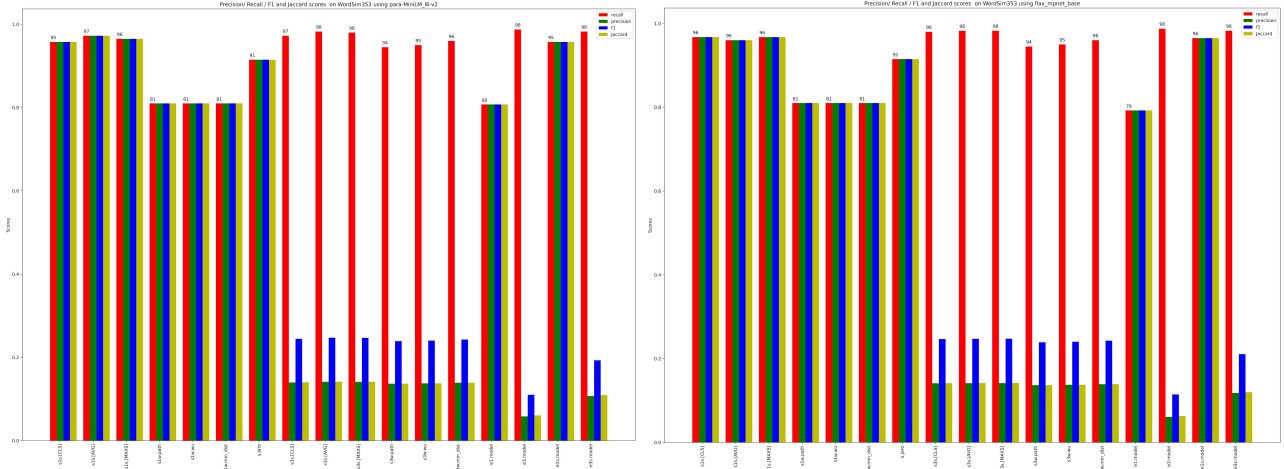

**Figure 12.** single methods result for *WordSim353*. Labels are those of Table 3a.

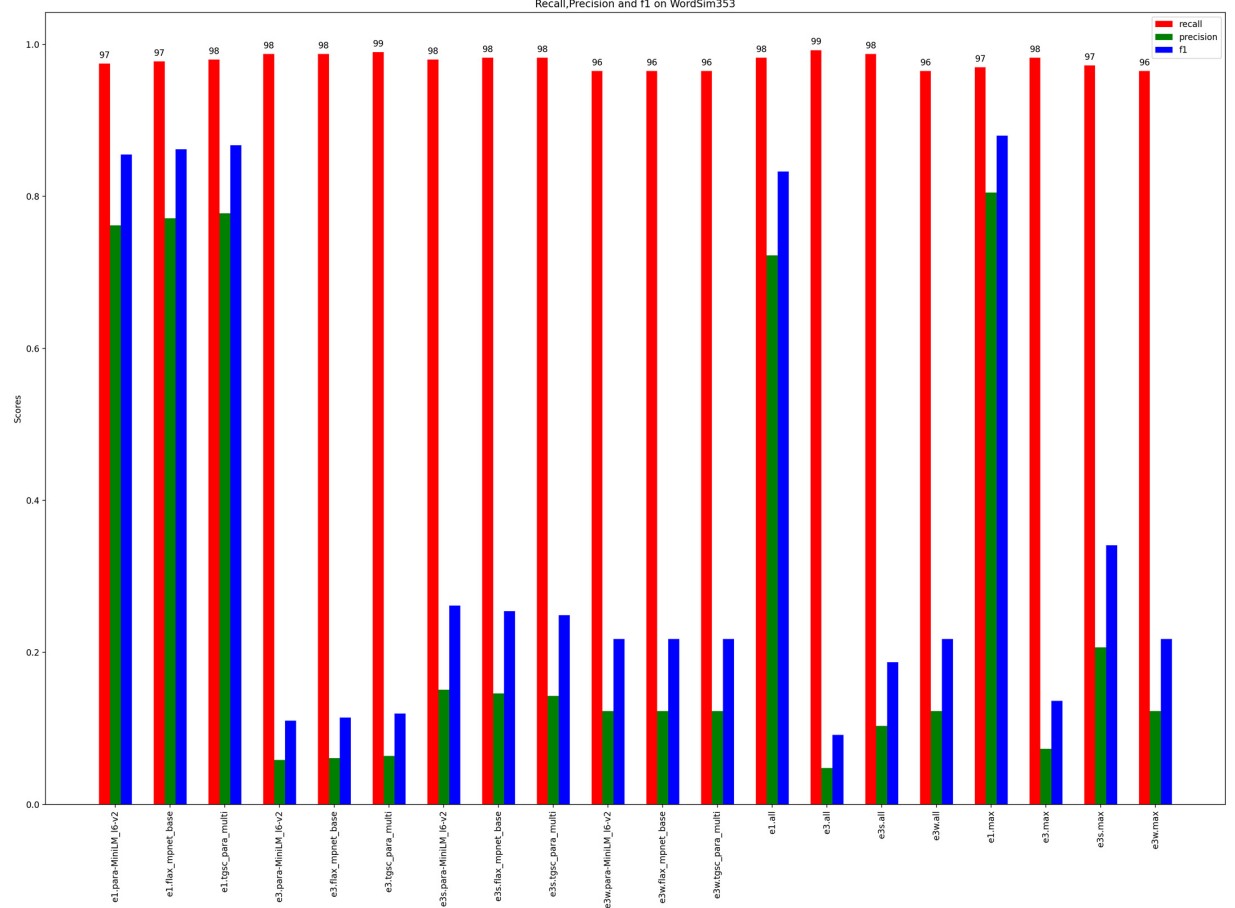

**Figure 13.** ensemble methods result for *WordSim353*. Labels are those of Table 3b.

High recall in the results indicates that the system is retrieving most of the relevant or correct values. In other words, it is finding many of the desired possible values. This is positive because our primary goal is to capture the most relevant information. The annotators have been specifically oriented to evaluate the similarity between tags that share semantic similarities or connections with the target tag: this explains why the system obtains results that may seem less precise according to traditional measures like F1-score and precision. The system captures the desired semantic connections, although it may be less precise in distinguishing between relevant and non-relevant values.

It is evident that when considering the top 3 results, the recall achieves its peak values, whereas the precision remains relatively low. This discrepancy arises because identifying the most similar tags expands the number of tags, leading to a decrease in precision. However, in the context of the study, this is not a concern since the objective is to retrieve all tags that are even remotely related, making it more inclined towards a "recall-oriented" approach rather than precision-focused.

The experimental setup has been implemented in Python 3.10, using standard packages like Numpy, Matplotlib, Pandas and other more specific ones for processing textual data such as NLTK, Gensim [40], and Sklearn, together with Pytorch, Transformers, Spacy and deep_translate and some experimental packages in GitHub. We used various pretrained models taken from HuggingFace and datasets from GitHub.

## 6. Conclusions and Future Works

In this article, we have presented an approach for evaluating the similarity between tags, both single and compound words, belonging to different datasets. This approach integrates various state-of-the-art methods and techniques. Specifically, we have tested methods that utilize pretrained language models, methods based on the hierarchical structure of WordNet, and string-based similarity measures used solely as reinforcement in majority voting mechanisms for ensemble models. These measures were subsequently composed using ensemble methods that showed the method's effectiveness. Experiments were conducted on various datasets characterized by challenges, such as multilingual datasets, Italian-only datasets, or datasets with highly specialized terms. In all the experiments, the best results were achieved by integrating the outcomes of different single similarity methods within each language model and globally across all models. The recall exceeded 90% for the top three results. Experimentation using the gold standard also produced optimal results.

The uniqueness of this approach lies in its fully unsupervised nature once the most suitable language models have been identified. It can be applied, almost without adaptation, to different datasets in real-world scenarios. Another characteristic of the method is its ability to adapt to multilingual datasets, thanks to automatic language detection. This feature is beneficial for both semantic models, as it aids in selecting the appropriate language model, and for WordNet-based models. Automatic language detection enables the method to handle different languages within the dataset seamlessly.

The presented pipeline offers several advantages by employing various criteria and integrating them to achieve better results than a solution that relies on a single method, albeit at the expense of response speed. In the future, efforts will be made to enhance this aspect.

Based on the progress and outcomes, the plan for future activities is to continue with the assessment process. This work requires additional evaluation in two critical areas: firstly, pertaining to the source datasets, test the method with far more data, and secondly, regarding the creation of ground truth data. This activity will require involving more evaluators with specialized backgrounds.

It is also intended to integrate it into QueryLab, from whose structure the need for the prototype emerged, to evaluate its functionality, usability, and usefulness fully. This integration will provide users additional flexibility in accessing and searching for information, enhancing their overall experience and enabling more effective queries.

**Author Contributions:** Conceptualization, I.G.; methodology, I.G.; software, I.G. and M.T.A.; valida­tion, I.G.; data curation, I.G and M.T.A.; writing—original draft preparation, I.G.; writing—review and editing, I.G. and M.T.A. All authors have read and agreed to the published version of the manuscript.

**Funding:** This research received no external funding.

**Institutional Review Board Statement:** Not applicable.

**Informed Consent Statement:** Not applicable.

**Data Availability Statement:** Some data are not publicly available due to research data ownership issues, while others can be found at the following address: http://arm.mi.imati.cnr.it/papers/bdcc2 023 (accessed on 15 September 2023).

**Conflicts of Interest:** The authors declare no conflict of interest.

## Appendix A

The results of the experimentation are shown in Figures 4–13.
**Mariage dataset**

**Table A1.** The single method results for *Mariage*, using the 'LLukas22/paraphrase-multilingual-mpnet-base-v2-embedding-all' pretrained language model, shown in Figure 4 on the left.

| Method | Recall | Precision | F1 | Dice | Jaccard |
|---|---|---|---|---|---|
| s1s.[CLS] | 0.346667 | 0.590909 | 0.436975 | 0.310606 | 0.215909 |
| s1s.[AVG] | 0.26506 | 0.5 | 0.346457 | 0.265152 | 0.185606 |
| s1s.[MAXS] | 0.394366 | 0.636364 | 0.486957 | 0.333333 | 0.231061 |
| s1w.path | 0.294872 | 0.522727 | 0.377049 | 0.261364 | 0.174242 |
| s1w.wu | 0.576271 | 0.772727 | 0.660194 | 0.397727 | 0.268939 |
| s1w.min_dist | 0.576271 | 0.772727 | 0.660194 | 0.397727 | 0.268939 |
| s.Jaro | 0.034483 | 0.090909 | 0.05 | 0.045455 | 0.030303 |
| s3s.[CLS] | 0.379032 | 0.152597 | 0.217593 | 0.216414 | 0.128923 |
| s3s.[AVG] | 0.370968 | 0.149351 | 0.212963 | 0.210732 | 0.125045 |
| s3s.[MAXS] | 0.354839 | 0.142857 | 0.203704 | 0.201641 | 0.119363 |
| s3w.path | 0.330645 | 0.138514 | 0.195238 | 0.187879 | 0.114313 |
| s3w.wu | 0.693548 | 0.296552 | 0.415459 | 0.393434 | 0.267181 |
| s3w.min_dist | 0.685484 | 0.293103 | 0.410628 | 0.389394 | 0.262716 |
| e1.model | 0.698113 | 0.840909 | 0.762887 | 0.431818 | 0.291667 |
| e3.model | 0.935484 | 0.127753 | 0.224806 | 0.226294 | 0.129223 |
| e1s.model | 0.394366 | 0.636364 | 0.486957 | 0.333333 | 0.231061 |
| e3s.model | 0.41129 | 0.141274 | 0.210309 | 0.211033 | 0.123785 |

**Table A2.** The single method results for *Mariage*, using the 'tgsc/sentence-transformers_paraphrase-multilingual-mpnet-base-v2' pretrained language model, shown in Figure 4 on the right.

| Method | Recall | Precision | F1 | Dice | Jaccard |
|---|---|---|---|---|---|
| s1s.[CLS] | 0.5 | 0.704545 | 0.584906 | 0.352273 | 0.234848 |
| s1s.[AVG] | 0.5 | 0.704545 | 0.584906 | 0.352273 | 0.234848 |
| s1s.[MAXS] | 0.568966 | 0.75 | 0.647059 | 0.375 | 0.25 |
| s1w.path | 0.294872 | 0.522727 | 0.377049 | 0.261364 | 0.174242 |
| s1w.wu | 0.576271 | 0.772727 | 0.660194 | 0.397727 | 0.268939 |
| s1w.min_dist | 0.576271 | 0.772727 | 0.660194 | 0.397727 | 0.268939 |
| s.Jaro | 0.034483 | 0.090909 | 0.05 | 0.045455 | 0.030303 |
| s3s.[CLS] | 0.5 | 0.201299 | 0.287037 | 0.283965 | 0.174874 |
| s3s.[AVG] | 0.491935 | 0.198052 | 0.282407 | 0.278914 | 0.173205 |
| s3s.[MAXS] | 0.491935 | 0.198052 | 0.282407 | 0.280556 | 0.172078 |
| s3w.path | 0.330645 | 0.138514 | 0.195238 | 0.187879 | 0.114313 |
| s3w.wu | 0.693548 | 0.296552 | 0.415459 | 0.393434 | 0.267181 |
| s3w.min_dist | 0.685484 | 0.293103 | 0.410628 | 0.389394 | 0.262716 |
| e1.model | 0.698113 | 0.840909 | 0.762887 | 0.431818 | 0.291667 |
| e3.model | 0.951613 | 0.133033 | 0.233432 | 0.235696 | 0.135516 |
| e1s.model | 0.568966 | 0.75 | 0.647059 | 0.375 | 0.25 |
| e3s.model | 0.580645 | 0.2 | 0.297521 | 0.296103 | 0.183816 |

**Table A3.** Ensemble methods result for *Mariage*, shown in Figure 5.

| Method | Recall | Precision | F1 |
|---|---|---|---|
| e1.llukas22_mpnet_base | 0.70297 | 0.669811 | 0.68599 |
| e1.tgsc_para_multi | 0.752577 | 0.715686 | 0.733668 |
| e3.llukas22_mpnet_base | 0.935484 | 0.127753 | 0.224806 |
| e3.tgsc_para_multi | 0.951613 | 0.133033 | 0.233432 |
| e3s.llukas22_mpnet_base | 0.362903 | 0.148026 | 0.21028 |
| e3s.tgsc_para_multi | 0.491935 | 0.203333 | 0.287736 |
| e3w.llukas22_mpnet_base | 0.754098 | 0.239583 | 0.363636 |
| e3w.tgsc_para_multi | 0.754098 | 0.239583 | 0.363636 |
| e1.all | 0.805556 | 0.644444 | 0.716049 |
| e3.all | 0.983871 | 0.122367 | 0.217663 |
| e3s.all | 0.620968 | 0.186893 | 0.287313 |
| e3w.all | 0.754098 | 0.239583 | 0.363636 |
| e1.max | 0.710843 | 0.766234 | 0.7375 |
| e3.max | 0.903226 | 0.140351 | 0.24295 |
| e3s.max | 0.233871 | 0.151042 | 0.183544 |
| e3w.max | 0.754098 | 0.239583 | 0.363636 |

## Wedding dataset

**Table A4.** Single method results for *Wedding*, using 'LLukas22/paraphrase-multilingual-mpnet-base-v2-embedding-all' pretrained language model, shown in Figure 6 on the right.

| Method | Recall | Precision | F1 | Dice | Jaccard |
|---|---|---|---|---|---|
| s1s.[CLS] | 0.65625 | 0.807692 | 0.724138 | 0.429487 | 0.294872 |
| s1s.[AVG] | 0.606061 | 0.769231 | 0.677966 | 0.403846 | 0.275641 |
| s1s.[MAXS] | 0.527778 | 0.730769 | 0.612903 | 0.391026 | 0.269231 |
| s1w.path | 0.414634 | 0.653846 | 0.507463 | 0.358974 | 0.25 |
| s1w.wu | 0.461538 | 0.692308 | 0.553846 | 0.384615 | 0.275641 |
| s1w.min_dist | 0.461538 | 0.692308 | 0.553846 | 0.384615 | 0.275641 |
| s.Jaro | 0.142857 | 0.307692 | 0.195122 | 0.166667 | 0.115385 |
| s3s.[CLS] | 0.59375 | 0.22619 | 0.327586 | 0.324306 | 0.207672 |
| s3s.[AVG] | 0.609375 | 0.232143 | 0.336207 | 0.332639 | 0.211806 |
| s3s.[MAXS] | 0.59375 | 0.22619 | 0.327586 | 0.324306 | 0.206019 |
| s3w.path | 0.4375 | 0.166667 | 0.241379 | 0.240741 | 0.145833 |
| s3w.wu | 0.671875 | 0.255952 | 0.37069 | 0.36875 | 0.241154 |
| s3w.min_dist | 0.671875 | 0.255952 | 0.37069 | 0.369676 | 0.236111 |
| e1.model | 0.59375 | 0.791667 | 0.678571 | 0.444444 | 0.319444 |
| e3.model | 0.984375 | 0.145833 | 0.254032 | 0.256155 | 0.148334 |
| e1s.model | 0.633333 | 0.791667 | 0.703704 | 0.423611 | 0.291667 |
| e3s.model | 0.65625 | 0.21 | 0.318182 | 0.322807 | 0.203907 |

**Table A5.** The single method results for *Wedding*, using the 'tgsc/sentence-transformers_paraphrase-multilingual-mpnet-base-v2' pretrained language model, shown in Figure 6 on the left.

| Method | Recall | Precision | F1 | Dice | Jaccard |
|---|---|---|---|---|---|
| s1s.[CLS] | 0.5 | 0.704545 | 0.584906 | 0.352273 | 0.234848 |
| s1s.[AVG] | 0.5 | 0.704545 | 0.584906 | 0.352273 | 0.234848 |
| s1s.[MAXS] | 0.568966 | 0.75 | 0.647059 | 0.375 | 0.25 |
| s1w.path | 0.294872 | 0.522727 | 0.377049 | 0.261364 | 0.174242 |
| s1w.wu | 0.576271 | 0.772727 | 0.660194 | 0.397727 | 0.268939 |
| s1w.min_dist | 0.576271 | 0.772727 | 0.660194 | 0.397727 | 0.268939 |
| s.Jaro | 0.034483 | 0.090909 | 0.05 | 0.045455 | 0.030303 |
| s3s.[CLS] | 0.5 | 0.201299 | 0.287037 | 0.283965 | 0.174874 |
| s3s.[AVG] | 0.491935 | 0.198052 | 0.282407 | 0.278914 | 0.173205 |
| s3s.[MAXS] | 0.491935 | 0.198052 | 0.282407 | 0.280556 | 0.172078 |
| s3w.path | 0.330645 | 0.138514 | 0.195238 | 0.187879 | 0.114313 |
| s3w.wu | 0.693548 | 0.296552 | 0.415459 | 0.393434 | 0.267181 |
| s3w.min_dist | 0.685484 | 0.293103 | 0.410628 | 0.389394 | 0.262716 |
| e1.model | 0.698113 | 0.840909 | 0.762887 | 0.431818 | 0.291667 |
| e3.model | 0.951613 | 0.133033 | 0.233432 | 0.235696 | 0.135516 |
| e1s.model | 0.568966 | 0.75 | 0.647059 | 0.375 | 0.25 |
| e3s.model | 0.580645 | 0.2 | 0.297521 | 0.296103 | 0.183816 |

**Table A6.** Ensemble methods result for *Wedding*, shown in Figure 7.

| Method | Recall | Precision | F1 |
|---|---|---|---|
| e1.llukas22_mpnet_base | 0.722222 | 0.661017 | 0.690265 |
| e1.tgsc_para_multi | 0.692308 | 0.610169 | 0.648649 |
| e3.llukas22_mpnet_base | 0.984375 | 0.145833 | 0.254032 |
| e3.tgsc_para_multi | 0.984375 | 0.146512 | 0.255061 |
| e3s.llukas22_mpnet_base | 0.578125 | 0.226994 | 0.325991 |
| e3s.tgsc_para_multi | 0.5625 | 0.219512 | 0.315789 |
| e3w.llukas22_mpnet_base | 0.703125 | 0.197368 | 0.308219 |
| e3w.tgsc_para_multi | 0.703125 | 0.197368 | 0.308219 |
| e1.all | 0.775862 | 0.616438 | 0.687023 |
| e3.all | 1 | 0.131959 | 0.233151 |
| e3s.all | 0.640625 | 0.199029 | 0.303704 |
| e3w.all | 0.703125 | 0.197368 | 0.308219 |
| e1.max | 0.625 | 0.666667 | 0.645161 |
| e3.max | 0.96875 | 0.164456 | 0.281179 |
| e3s.max | 0.5 | 0.264463 | 0.345946 |
| e3w.max | 0.703125 | 0.197368 | 0.308219 |

## ICH tags dataset

**Table A7.** Single methods result for *ICH Tags*, using the 'sentence-transformers/paraphrase-MiniLM-L6-v2' pretrained language model, shown in Figure 8 on the left.

| Method | Recall | Precision | F1 | Dice | Jaccard |
|---|---|---|---|---|---|
| s1s.[CLS] | 0.460177 | 0.693333 | 0.553191 | 0.351111 | 0.235556 |
| s1s.[AVG] | 0.447368 | 0.68 | 0.539683 | 0.342222 | 0.228889 |
| s1s.[MAXS] | 0.477477 | 0.706667 | 0.569892 | 0.357778 | 0.24 |
| s1w.path | 0.320611 | 0.56 | 0.407767 | 0.28 | 0.186667 |
| s1w.wu | 0.504673 | 0.72 | 0.593407 | 0.36 | 0.24 |
| s1w.min_dist | 0.504673 | 0.72 | 0.593407 | 0.36 | 0.24 |
| s.Jaro | 0.121387 | 0.28 | 0.169355 | 0.14 | 0.093333 |
| s3s.[CLS] | 0.466346 | 0.19246 | 0.272472 | 0.271373 | 0.166336 |
| s3s.[AVG] | 0.485577 | 0.200397 | 0.283708 | 0.282793 | 0.173473 |
| s3s.[MAXS] | 0.480769 | 0.198413 | 0.280899 | 0.279707 | 0.172564 |
| s3w.path | 0.346154 | 0.142857 | 0.202247 | 0.200617 | 0.121142 |
| s3w.wu | 0.677885 | 0.279762 | 0.396067 | 0.392593 | 0.260444 |
| s3w.min_dist | 0.644231 | 0.265873 | 0.376404 | 0.37284 | 0.244819 |
| e1.model | 0.59375 | 0.791667 | 0.678571 | 0.400463 | 0.268519 |
| e3.model | 0.961538 | 0.135227 | 0.237107 | 0.242837 | 0.139389 |
| e1s.model | 0.454545 | 0.694444 | 0.549451 | 0.351852 | 0.236111 |
| e3s.model | 0.576923 | 0.169492 | 0.262009 | 0.269189 | 0.162208 |

**Table A8.** Single methods result for *ICH Tags*, using the 'flax-sentence-embeddings/all_datasets_v3_mpnet-base' pretrained language model, shown in Figure 8 on the right.

| Method | Recall | Precision | F1 | Dice | Jaccard |
|---|---|---|---|---|---|
| s1s.[CLS] | 0.346154 | 0.6 | 0.439024 | 0.313333 | 0.215556 |
| s1s.[AVG] | 0.390244 | 0.64 | 0.484848 | 0.331111 | 0.226667 |
| s1s.[MAXS] | 0.341085 | 0.586667 | 0.431373 | 0.297778 | 0.2 |
| s1w.path | 0.320611 | 0.56 | 0.407767 | 0.28 | 0.186667 |
| s1w.wu | 0.504673 | 0.72 | 0.593407 | 0.36 | 0.24 |
| s1w.min_dist | 0.504673 | 0.72 | 0.593407 | 0.36 | 0.24 |
| s.Jaro | 0.121387 | 0.28 | 0.169355 | 0.14 | 0.093333 |
| s3s.[CLS] | 0.389423 | 0.160714 | 0.227528 | 0.226929 | 0.135362 |
| s3s.[AVG] | 0.432692 | 0.178571 | 0.252809 | 0.252238 | 0.15264 |
| s3s.[MAXS] | 0.442308 | 0.18254 | 0.258427 | 0.257485 | 0.157132 |
| s3w.path | 0.346154 | 0.142857 | 0.202247 | 0.200617 | 0.121142 |
| s3w.wu | 0.677885 | 0.279762 | 0.396067 | 0.392593 | 0.260444 |
| s3w.min_dist | 0.644231 | 0.265873 | 0.376404 | 0.37284 | 0.244819 |
| e1.model | 0.583333 | 0.777778 | 0.666667 | 0.388889 | 0.259259 |
| e3.model | 0.951923 | 0.139437 | 0.243243 | 0.247863 | 0.142883 |
| e1s.model | 0.333333 | 0.583333 | 0.424242 | 0.296296 | 0.199074 |
| e3s.model | 0.480769 | 0.160514 | 0.240674 | 0.244501 | 0.146704 |

**Table A9.** Single methods result for *ICH Tags*, using the 'tgsc/sentence-transformers_paraphrase-multilingual-mpnet-base-v2' pretrained language model.

| Method | Recall | Precision | F1 | Dice | Jaccard |
|---|---|---|---|---|---|
| s1s.[CLS] | 0.390244 | 0.64 | 0.484848 | 0.331111 | 0.226667 |
| s1s.[AVG] | 0.518519 | 0.746667 | 0.612022 | 0.386667 | 0.264444 |
| s1s.[MAXS] | 0.420168 | 0.666667 | 0.515464 | 0.344444 | 0.235556 |
| s1w.path | 0.320611 | 0.56 | 0.407767 | 0.28 | 0.186667 |
| s1w.wu | 0.504673 | 0.72 | 0.593407 | 0.36 | 0.24 |
| s1w.min_dist | 0.504673 | 0.72 | 0.593407 | 0.36 | 0.24 |
| s.Jaro | 0.121387 | 0.28 | 0.169355 | 0.14 | 0.093333 |
| s3s.[CLS] | 0.480769 | 0.198413 | 0.280899 | 0.280324 | 0.171682 |
| s3s.[AVG] | 0.504808 | 0.208333 | 0.294944 | 0.294213 | 0.182429 |
| s3s.[MAXS] | 0.466346 | 0.19246 | 0.272472 | 0.271373 | 0.167879 |
| s3w.path | 0.346154 | 0.142857 | 0.202247 | 0.200617 | 0.121142 |
| s3w.wu | 0.677885 | 0.279762 | 0.396067 | 0.392593 | 0.260444 |
| s3w.min_dist | 0.644231 | 0.265873 | 0.376404 | 0.37284 | 0.244819 |
| e1.model | 0.519608 | 0.736111 | 0.609195 | 0.368056 | 0.24537 |
| e3.model | 0.942308 | 0.142029 | 0.246851 | 0.250538 | 0.144767 |
| e1s.model | 0.413793 | 0.666667 | 0.510638 | 0.344907 | 0.236111 |
| e3s.model | 0.528846 | 0.190972 | 0.280612 | 0.28209 | 0.172596 |

**Table A10.** Ensemble methods result for *ICH Tags*, shown in Figure 9.

| Method | Recall | Precision | F1 |
|---|---|---|---|
| e1.para-MiniLM_l6-v2 | 0.672414 | 0.62234 | 0.646409 |
| e1.flax_mpnet_base | 0.620112 | 0.593583 | 0.606557 |
| e1.tgsc_para_multi | 0.625731 | 0.601124 | 0.613181 |
| e3.para-MiniLM_l6-v2 | 0.961538 | 0.135227 | 0.237107 |
| e3.flax_mpnet_base | 0.951923 | 0.139437 | 0.243243 |
| e3.tgsc_para_multi | 0.942308 | 0.142029 | 0.246851 |
| e3s.para-MiniLM_l6-v2 | 0.485577 | 0.221491 | 0.304217 |
| e3s.flax_mpnet_base | 0.423077 | 0.183716 | 0.256186 |
| e3s.tgsc_para_multi | 0.490385 | 0.204819 | 0.288952 |
| e3w.para-MiniLM_l6-v2 | 0.677885 | 0.206745 | 0.316854 |
| e3w.flax_mpnet_base | 0.677885 | 0.206745 | 0.316854 |
| e3w.tgsc_para_multi | 0.677885 | 0.206745 | 0.316854 |
| e1.all | 0.771277 | 0.516014 | 0.618337 |
| e3.all | 0.985577 | 0.113135 | 0.20297 |
| e3s.all | 0.615385 | 0.175824 | 0.273504 |
| e3w.all | 0.677885 | 0.206745 | 0.316854 |
| e1.max | 0.611511 | 0.714286 | 0.658915 |
| e3.max | 0.889423 | 0.157046 | 0.266955 |
| e3s.max | 0.307692 | 0.20915 | 0.249027 |
| e3w.max | 0.677885 | 0.206745 | 0.316854 |

## Cook_IT dataset

**Table A11.** The single method results for *Cook_IT*, using the 'tgsc/sentence-transformers_paraphrase-multilingual-mpnet-base-v2' pretrained language model, shown in Figure 10 on the left.

| Method | Recall | Precision | F1 | Dice | Jaccard |
|---|---|---|---|---|---|
| s1s.[CLS] | 0.275862 | 0.380952 | 0.32 | 0.261905 | 0.214286 |
| s1s.[AVG] | 0.333333 | 0.428571 | 0.375 | 0.285714 | 0.230159 |
| s1s.[MAXS] | 0.275862 | 0.380952 | 0.32 | 0.261905 | 0.214286 |
| s1w.path | 0.321429 | 0.428571 | 0.367347 | 0.301587 | 0.253968 |
| s1w.wu | 0.37037 | 0.47619 | 0.416667 | 0.333333 | 0.277778 |
| s1w.min_dist | 0.37037 | 0.47619 | 0.416667 | 0.333333 | 0.277778 |
| s.Jaro | 0.258065 | 0.380952 | 0.307692 | 0.285714 | 0.246032 |
| s3s.[CLS] | 0.289474 | 0.079137 | 0.124294 | 0.124565 | 0.072184 |
| s3s.[AVG] | 0.315789 | 0.084507 | 0.133333 | 0.130385 | 0.075113 |
| s3s.[MAXS] | 0.289474 | 0.078571 | 0.123596 | 0.122184 | 0.070673 |
| s3w.path | 0.421053 | 0.153846 | 0.225352 | 0.175737 | 0.111111 |
| s3w.wu | 0.5 | 0.197917 | 0.283582 | 0.243764 | 0.166667 |
| s3w.min_dist | 0.5 | 0.2 | 0.285714 | 0.245881 | 0.168367 |
| e1.model | 0.423077 | 0.52381 | 0.468085 | 0.365079 | 0.301587 |
| e3.model | 0.631579 | 0.086331 | 0.151899 | 0.140513 | 0.081068 |
| e1s.model | 0.333333 | 0.428571 | 0.375 | 0.285714 | 0.230159 |
| e3s.model | 0.315789 | 0.07362 | 0.119403 | 0.122151 | 0.069766 |

**Table A12.** The single method results for *Cook_IT*, using the 'nickprock/sentence-bert-base-italian-xxl-uncased' pretrained language model, shown in Figure 10 on the right.

| Method | Recall | Precision | F1 | Dice | Jaccard |
|---|---|---|---|---|---|
| s1s.[CLS] | 0.428571 | 0.571429 | 0.489796 | 0.436508 | 0.380952 |
| s1s.[AVG] | 0.392857 | 0.52381 | 0.44898 | 0.388889 | 0.333333 |
| s1s.[MAXS] | 0.423077 | 0.52381 | 0.468085 | 0.365079 | 0.301587 |
| s1w.path | 0.321429 | 0.428571 | 0.367347 | 0.301587 | 0.253968 |
| s1w.wu | 0.37037 | 0.47619 | 0.416667 | 0.333333 | 0.277778 |
| s1w.min_dist | 0.37037 | 0.47619 | 0.416667 | 0.333333 | 0.277778 |
| s.Jaro | 0.258065 | 0.380952 | 0.307692 | 0.285714 | 0.246032 |
| s3s.[CLS] | 0.578947 | 0.152778 | 0.241758 | 0.241081 | 0.145597 |
| s3s.[AVG] | 0.578947 | 0.153846 | 0.243094 | 0.241081 | 0.145597 |
| s3s.[MAXS] | 0.605263 | 0.159722 | 0.252747 | 0.250605 | 0.150888 |
| s3w.path | 0.421053 | 0.153846 | 0.225352 | 0.175737 | 0.111111 |
| s3w.wu | 0.5 | 0.197917 | 0.283582 | 0.243764 | 0.166667 |
| s3w.min_dist | 0.5 | 0.2 | 0.285714 | 0.245881 | 0.168367 |
| e1.model | 0.461538 | 0.571429 | 0.510638 | 0.412698 | 0.349206 |
| e3.model | 0.789474 | 0.109091 | 0.191693 | 0.190999 | 0.110799 |
| e1s.model | 0.461538 | 0.571429 | 0.510638 | 0.412698 | 0.349206 |
| e3s.model | 0.605263 | 0.138554 | 0.22549 | 0.229733 | 0.136675 |

**Table A13.** Ensemble methods result for *Cook_IT*, shown in Figure 11.

| Method | Recall | Precision | F1 |
|---|---|---|---|
| e1.nickprock_ita | 0.566667 | 0.395349 | 0.465753 |
| e1.tgsc_para_multi | 0.5 | 0.365854 | 0.422535 |
| e3.nickprock_ita | 0.789474 | 0.109091 | 0.191693 |
| e3.tgsc_para_multi | 0.631579 | 0.086331 | 0.151899 |
| e3s.nickprock_ita | 0.578947 | 0.157143 | 0.247191 |
| e3s.tgsc_para_multi | 0.315789 | 0.086331 | 0.135593 |
| e3w.nickprock_ita | 0.714286 | 0.183486 | 0.291971 |
| e3w.tgsc_para_multi | 0.714286 | 0.183486 | 0.291971 |
| e1.all | 0.645161 | 0.322581 | 0.430108 |
| e3.all | 0.868421 | 0.092697 | 0.167513 |
| e3s.all | 0.657895 | 0.123153 | 0.207469 |
| e3w.all | 0.714286 | 0.183486 | 0.291971 |
| e1.max | 0.428571 | 0.363636 | 0.393443 |
| e3.max | 0.552632 | 0.106599 | 0.178723 |
| e3s.max | 0.236842 | 0.118421 | 0.157895 |
| e3w.max | 0.714286 | 0.183486 | 0.291971 |

**WordSim353 dataset**

**Table A14.** Single methods result for *WordSim353*, using 'sentence-transformers/paraphrase-MiniLM-L6-v2' pretrained language model, shown in Figure 12, on the left.

| Method | Recall | Precision | F1 | Dice | Jaccard |
|---|---|---|---|---|---|
| **s1s.[CLS]** | 0.9575 | 0.9575 | 0.9575 | 0.9575 | 0.9575 |
| **s1s.[AVG]** | 0.9725 | 0.9725 | 0.9725 | 0.9725 | 0.9725 |
| **s1s.[MAXS]** | 0.965 | 0.965 | 0.965 | 0.965 | 0.965 |
| **s1w.path** | 0.81 | 0.81 | 0.81 | 0.81 | 0.81 |
| **s1w.wu** | 0.81 | 0.81 | 0.81 | 0.81 | 0.81 |
| **s1w.min_dist** | 0.81 | 0.81 | 0.81 | 0.81 | 0.81 |
| **s.Jaro** | 0.915 | 0.915 | 0.915 | 0.915 | 0.915 |
| **s3s.[CLS]** | 0.9725 | 0.139777 | 0.244423 | 0.244643 | 0.13994 |
| **s3s.[AVG]** | 0.9825 | 0.141367 | 0.24717 | 0.247321 | 0.141488 |
| **s3s.[MAXS]** | 0.98 | 0.14116 | 0.246774 | 0.246964 | 0.14131 |
| **s3w.path** | 0.945 | 0.136758 | 0.238938 | 0.23881 | 0.136714 |
| **s3w.wu** | 0.95 | 0.137532 | 0.240278 | 0.240149 | 0.137488 |
| **s3w.min_dist** | 0.96 | 0.138929 | 0.242731 | 0.24256 | 0.138857 |
| **e1.model** | 0.8075 | 0.8075 | 0.8075 | 0.8075 | 0.8075 |
| **e3.model** | 0.9875 | 0.058277 | 0.110059 | 0.1137 | 0.060448 |
| **e1s.model** | 0.9575 | 0.9575 | 0.9575 | 0.9575 | 0.9575 |
| **e3s.model** | 0.9825 | 0.107026 | 0.193026 | 0.196602 | 0.109461 |

**Table A15.** Single methods result for *WordSim353*, using the 'flax-sentence-embeddings/all_datasets_v3_mpnet-base' pretrained language model, shown in Figure 12 on the right.

| Method | Recall | Precision | F1 | Dice | Jaccard |
|---|---|---|---|---|---|
| **s1s.[CLS]** | 0.9675 | 0.9675 | 0.9675 | 0.9675 | 0.9675 |
| **s1s.[AVG]** | 0.96 | 0.96 | 0.96 | 0.96 | 0.96 |
| **s1s.[MAXS]** | 0.9675 | 0.9675 | 0.9675 | 0.9675 | 0.9675 |
| **s1w.path** | 0.81 | 0.81 | 0.81 | 0.81 | 0.81 |
| **s1w.wu** | 0.81 | 0.81 | 0.81 | 0.81 | 0.81 |
| **s1w.min_dist** | 0.81 | 0.81 | 0.81 | 0.81 | 0.81 |
| **s.Jaro** | 0.915 | 0.915 | 0.915 | 0.915 | 0.915 |
| **s3s.[CLS]** | 0.98 | 0.141109 | 0.246696 | 0.246964 | 0.14131 |
| **s3s.[AVG]** | 0.9825 | 0.141316 | 0.247092 | 0.247321 | 0.141488 |
| **s3s.[MAXS]** | 0.9825 | 0.14152 | 0.247403 | 0.247679 | 0.141726 |
| **s3w.path** | 0.945 | 0.136758 | 0.238938 | 0.23881 | 0.136714 |
| **s3w.wu** | 0.95 | 0.137532 | 0.240278 | 0.240149 | 0.137488 |
| **s3w.min_dist** | 0.96 | 0.138929 | 0.242731 | 0.24256 | 0.138857 |
| **e1.model** | 0.7925 | 0.7925 | 0.7925 | 0.7925 | 0.7925 |
| **e3.model** | 0.9875 | 0.060713 | 0.114393 | 0.117832 | 0.062778 |
| **e1s.model** | 0.965 | 0.965 | 0.965 | 0.965 | 0.965 |
| **e3s.model** | 0.9825 | 0.117947 | 0.210611 | 0.213335 | 0.119866 |

**Table A16.** single methods result for *WordSim353*, using 'tgsc/sentence-transformers_paraphrase-multilingual-mpnet-base-v2' pretrained language model.

| Method | Recall | Precision | F1 | Dice | Jaccard |
| --- | --- | --- | --- | --- | --- |
| **s1s.[CLS]** | 0.97 | 0.97 | 0.97 | 0.97 | 0.97 |
| **s1s.[AVG]** | 0.97 | 0.97 | 0.97 | 0.97 | 0.97 |
| **s1s.[MAXS]** | 0.9675 | 0.9675 | 0.9675 | 0.9675 | 0.9675 |
| **s1w.path** | 0.81 | 0.81 | 0.81 | 0.81 | 0.81 |
| **s1w.wu** | 0.81 | 0.81 | 0.81 | 0.81 | 0.81 |
| **s1w.min_dist** | 0.81 | 0.81 | 0.81 | 0.81 | 0.81 |
| **s.Jaro** | 0.915 | 0.915 | 0.915 | 0.915 | 0.915 |
| **s3s.[CLS]** | 0.9825 | 0.141469 | 0.247325 | 0.247589 | 0.141667 |
| **s3s.[AVG]** | 0.9825 | 0.141622 | 0.247559 | 0.247857 | 0.141845 |
| **s3s.[MAXS]** | 0.9825 | 0.141469 | 0.247325 | 0.247589 | 0.141667 |
| **s3w.path** | 0.945 | 0.136758 | 0.238938 | 0.23881 | 0.136714 |
| **s3w.wu** | 0.95 | 0.137532 | 0.240278 | 0.240149 | 0.137488 |
| **s3w.min_dist** | 0.96 | 0.138929 | 0.242731 | 0.24256 | 0.138857 |
| **e1.model** | 0.8175 | 0.8175 | 0.8175 | 0.8175 | 0.8175 |
| **e3.model** | 0.99 | 0.063604 | 0.119529 | 0.123139 | 0.065788 |
| **e1s.model** | 0.9675 | 0.9675 | 0.9675 | 0.9675 | 0.9675 |
| **e3s.model** | 0.9825 | 0.12793 | 0.226382 | 0.22864 | 0.129532 |

**Table A17.** Ensemble methods result for *WordSim353*, shown in Figure 13.

| Method | Recall | Precision | F1 |
| --- | --- | --- | --- |
| **e1.para-MiniLM_l6-v2** | 0.975 | 0.761719 | 0.855263 |
| **e1.flax_mpnet_base** | 0.9775 | 0.771203 | 0.862183 |
| **e1.tgsc_para_multi** | 0.98 | 0.777778 | 0.867257 |
| **e3.para-MiniLM_l6-v2** | 0.9875 | 0.058277 | 0.110059 |
| **e3.flax_mpnet_base** | 0.9875 | 0.060713 | 0.114393 |
| **e3.tgsc_para_multi** | 0.99 | 0.063604 | 0.119529 |
| **e3s.para-MiniLM_l6-v2** | 0.98 | 0.151002 | 0.261682 |
| **e3s.flax_mpnet_base** | 0.9825 | 0.146151 | 0.254451 |
| **e3s.tgsc_para_multi** | 0.9825 | 0.14265 | 0.249128 |
| **e3w.para-MiniLM_l6-v2** | 0.964912 | 0.122651 | 0.217637 |
| **e3w.flax_mpnet_base** | 0.964912 | 0.122651 | 0.217637 |
| **e3w.tgsc_para_multi** | 0.964912 | 0.122651 | 0.217637 |
| **e1.all** | 0.9825 | 0.722426 | 0.832627 |
| **e3.all** | 0.9925 | 0.047953 | 0.091485 |
| **e3s.all** | 0.9875 | 0.103268 | 0.186982 |
| **e3w.all** | 0.964912 | 0.122651 | 0.217637 |
| **e1.max** | 0.97 | 0.804979 | 0.879819 |
| **e3.max** | 0.9825 | 0.07328 | 0.136387 |
| **e3s.max** | 0.9725 | 0.206695 | 0.340929 |
| **e3w.max** | 0.964912 | 0.122651 | 0.217637 |

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
