# Peer review of "Ensemble-Based Short Text Similarity: An Easy Approach for Multilingual Datasets Using Transformers and WordNet in Real-World Scenarios"

_2504-2289, doi:10.3390/bdcc7040158_

Round 1

Reviewer 1 Report

The authors design an unsupervised approach for evaluating semantic similarity of texts in multilingual contexts. It proposes a novel approach for integrating multiple languages, as well as methods for finding word similarity.The paper, however, lacks clarity in its writing.The approach, experimentation, and results sections are confusing.It should be divided into subsections and made more concise and clear.Figures are difficult to read.

Additional comments:

Lines #22-30. One paragraph may be sufficient.

Lines # 25-30. References to relevant sources or related research should be cited.

Line #66. The QueryLab portal should be cited

Lines # 69-73. When referencing, please include section numbers.

Line #89. When referring to a research, please include the author's name (for ex. X. et.al). The same should be applied to all references.

Lines #100. It is important to cite sources when you refer to research/methods.

Lines #259-260. Please cite your sources

Line # 278. Cite the related works for different voting approaches.

The dataset section should be included before the approach section.

The approach section is confusing.It is difficult to gain a general understanding of the approach.It is important to improve the writing, make it more concise, and make it clearer.The approach subsections should be referred to the design diagram.

It is difficult to read the figure labels and axis names.

Please break down the experiment/result into subsections and describe the different evaluations that were performed. Results are difficult to understand. For a better understanding of the methods used in experiments, please refer to the respective approach section.

Writing should be improved. Please see above comments. 

Author Response

Ensemble-based Short Text Similarity: An Easy Approach for Multilingual Datasets using Transformers and WordNet in Real-world Scenarios

Isabella Gagliardi*, Maria Teresa Artese

Manuscript ID: BDCC-2504601

We are thankful to the editor and anonymous reviewers for reviewing our article and for the valuable comments and suggestions.

The current version of the paper has been carefully and extensively revised following the reviewers’ suggestions. We have taken into account the reviewers comments and we have added paragraphs and new information, to respond to their requests as described in detail below.

Reviewer #1

The authors design an unsupervised approach for evaluating semantic similarity of texts in multilingual contexts. It proposes a novel approach for integrating multiple languages, as well as methods for finding word similarity. The paper, however, lacks clarity in its writing. The approach, experimentation, and results sections are confusing. It should be divided into subsections and made more concise and clear. Figures are difficult to read.

Additional comments:

Lines #22-30. One paragraph may be sufficient.

We thank the reviewer for their valuable feedback. In response, we have revisit the initial two sentences, carefully streamlining them to achieve greater conciseness and brevity.

Lines # 25-30. References to relevant sources or related research should be cited. 

Line #66. The QueryLab portal should be cited 

Lines # 69-73. When referencing, please include section numbers.

Lines #100. It is important to cite sources when you refer to research/methods.

Lines #259-260. Please cite your sources

Line # 278. Cite the related works for different voting approaches.

We sincerely appreciate the reviewer's constructive feedback. In response, we have meticulously incorporated relevant sources and works into all the lines highlighted by the reviewer. 

Line #89. When referring to a research, please include the author's name (for ex. X. et.al). The same should be applied to all references.

Regarding the bibliographic citations within the text and in the references section, we have adhered to the journal's guidelines, which require them to be presented within square brackets in the order of their appearance in the text.

The dataset section should be included before the approach section.

The article's structure involves describing the methodological part within the approach, and then, in the experimental section, detailing the data, implementation, and obtained results, along with some critical comments. Therefore, we have placed the description of the datasets after the approach section.

The approach section is confusing. It is difficult to gain a general understanding of the approach. It is important to improve the writing, make it more concise, and make it clearer. The approach subsections should be referred to the design diagram.

We thank the reviewer for his helpful notes. We have revised and included in the paper all the suggested changes.

It is difficult to read the figure labels and axis names.

Indeed, the figures and, in particular, the labels are challenging to read. To enhance comprehension, we have provided more detailed information in tables 3a and 3b. Additionally, we have included tables with the results of various conducted experiments as an appendix.

Please break down the experiment/result into subsections and describe the different evaluations that were performed. Results are difficult to understand. For a better understanding of the methods used in experiments, please refer to the respective approach section.

We sincerely appreciate the reviewer's valuable feedback, which was instrumental in improving our paper. We have thoroughly reviewed and incorporated all the suggested changes in the manuscript. We have added references to the various sections of the article to improve understanding of the approach and the results obtained.

Reviewer 2 Report

The authors proposed a pipeline to identify similarities in 1- or n-gram tags, to evaluate and compare different pre-trained language models, and to define integrated methods to overcome limitations. Tests to validate the approach have been conducted using the QueryLab portal, a search engine for cultural heritage archives, to evaluate the proposed pipeline.

  Major comments 1. The evaluation metrics seems highly fluctuating, it's better to provide the results in a table. 2. The authors claims that the proposed work focuses on the objective to improve the recall. Using the ensemble method they achieved the recall in 90 percentage as well. But with a below average f score how can a high recall system can be claimed to be robust 3. In the introduction section the authors described Word2Vec, Glove and Fasttext approaches. These systems reported good performance in sentence similarity tasks. What is the purgative of not including those methods in evaluation 4. Transformers models and ensemble methods seems performing well, but the time and programming effort and deployment for generalizability are time consuming and complex. Is it worth developing a simple and light weight model or at least find the performance of those methods. 5. Since human annotation is performed, please report the inter rater reliability IRR 6. Provide explanation of specific and global input/output in figure 1 7. In table 2 various corpus used for the model evaluation is reported. How many of the tags were used from each corpus  8. For machine learning models in general the train and test sets are split into 70/30 or 80/20 ratio. What is the ratio used the current study. Why there is no development set for hyper parameter optimization 9. Explain this: Genism [Error! Reference source not found.] Why there is no reference for a widely used NLP package   Over all comments The authors reported various methods to identify similarities in 1- or n-gram tags. The task is relevant, and the models were applied to a real world tasks. The results are interesting and reportable. The manuscript can be accepted with minor revisions.

Author Response

Ensemble-based Short Text Similarity: An Easy Approach for Multilingual Datasets using Transformers and WordNet in Real-world Scenarios

Isabella Gagliardi*, Maria Teresa Artese

Manuscript ID: BDCC-2504601

We are thankful to the editor and anonymous reviewers for reviewing our article and for the valuable comments and suggestions.

The current version of the paper has been carefully and extensively revised following the reviewers’ suggestions. We have taken into account the reviewers comments and we have added paragraphs and new information, to respond to their requests as described in detail below.

Line numbers refer to the file of the revised paper in pdf format.

Reviewer #2

The authors proposed a pipeline to identify similarities in 1- or n-gram tags, to evaluate and compare different pre-trained language models, and to define integrated methods to overcome limitations. Tests to validate the approach have been conducted using the QueryLab portal, a search engine for cultural heritage archives, to evaluate the proposed pipeline.

  Major comments

  1. The evaluation metrics seems highly fluctuating, it's better to provide the results in a table.

We thank the reviewer for their valuable feedback. The numerical data derived from our experimentation, as depicted in the graphs, has been incorporated in the form of tables within the appendix A.

  1. The authors claims that the proposed work focuses on the objective to improve the recall. Using the ensemble method they achieved the recall in 90 percentage as well. But with a below average f score how can a high recall system can be claimed to be robust

This experimentation is conducted with the objective of retrieving terms that are similar or somewhat related. Consequently, both the terms deemed most similar and the three terms with the highest degrees of similarity, as identified by the individual models and their amalgamation, have been utilized for the evaluation. In order to enhance clarity and assist the reader in comprehending the results, a brief paragraph has been thoughtfully included (lines 591-597).

  1. In the introduction section the authors described Word2Vec, Glove and Fasttext approaches. These systems reported good performance in sentence similarity tasks. What is the purgative of not including those methods in evaluation

In our previous experiments and research, we frequently used word embeddings like Word2Vec or GloVe. In lines 36 - 46, we have provided an explanation for why, in this particular work, we opted to emphasize the use of transformer models such as BERT or other pre-trained language models.

  1. Transformers models and ensemble methods seems performing well, but the time and programming effort and deployment for generalizability are time consuming and complex. Is it worth developing a simple and light weight model or at least find the performance of those methods.

While it's true that testing different models and integrating them adds complexity to the approach outlined in this article, making it somewhat less efficient than a simpler solution, we have elaborated on this matter in the section spanning lines 625 to 627.

  1. Since human annotation is performed, please report the inter rater reliability IRR

We did engage with a few users and cultural heritage experts to assess the results: their participation was limited in number, and as a result, we have not provided an Inter-Rater Reliability (IRR) measurement as indicated in lines 474-475. We have plans to conduct a more extensive evaluation in upcoming work, as mentioned in lines 628-635.

  1. Provide explanation of specific and global input/output in figure 1

We have thoughtfully included the input and output descriptions for Figure 1 in the section indicated by lines.

  1. In table 2 various corpus used for the model evaluation is reported. How many of the tags were used from each corpus 

We have included the count of elements utilized for these experiments in the paragraph 5.1.

  1. For machine learning models in general the train and test sets are split into 70/30 or 80/20 ratio. What is the ratio used the current study. Why there is no development set for hyper parameter optimization

In this experiment, only pretrained models were utilized, given the insufficient size of the corpus, as detailed in lines 53-54.

  1. Explain this: Genism [Error! Reference source not found.] Why there is no reference for a widely used NLP package  

I am uncertain about the circumstances that led to the misplacement of the bibliographic reference. I would like to inform you that it has been reinstated.

Over all comments The authors reported various methods to identify similarities in 1- or n-gram tags. The task is relevant, and the models were applied to a real world tasks. The results are interesting and reportable. The manuscript can be accepted with minor revisions.

Expressing our sincere gratitude for your invaluable input and constructive feedback, which have played an instrumental role in enhancing the quality and overall refinement of our paper.

Round 2

Reviewer 1 Report

The authors have modified the manuscript according to my feedback.